# Hybridizing Bayesian and variational data assimilation for robust high-resolution hydrologic forecasting

Felipe Hernández, Xu Liang

Civil and Environmental Engineering Department, University of Pittsburgh, Pittsburgh, 15213, United States of America

*Correspondence to*: Xu Liang (xuliang@pitt.edu)

**Abstract.** The success of real-time estimation and forecasting applications based on geophysical models has been possible thanks to the two main existing frameworks for the determination of the models' initial conditions: Bayesian data assimilation and variational data assimilation. However, while there have been efforts to unify these two paradigms, existing attempts struggle to fully leverage the advantages of both in order to face the challenges posed by modern high-resolution models—
mainly related to model indeterminacy and steep computational requirements. In this article we introduce a hybrid algorithm called OPTIMISTS (Optimized PareTo Inverse Modeling through Integrated STochastic Search) which is targeted at non-linear high-resolution problems and that brings together ideas from particle filters, 4-dimensional variational methods, evolutionary Pareto optimization, and kernel density estimation in a unique way. Streamflow forecasting experiments were conducted to test which specific configurations of OPTIMISTS led to higher predictive accuracy. The experiments were
conducted on two watersheds: the Blue River (low-resolution) using the VIC (Variable Infiltration Capacity) model and the Indiantown Run (high-resolution) using the DHSVM (Distributed Hydrology Soil Vegetation Model). By selecting kernel-based non-parametric sampling, non-sequential evaluation of candidate particles, and through the multi-objective minimization of departures from the streamflow observations and from the background states, OPTIMISTS was shown to efficiently produce probabilistic forecasts with comparable accuracy to that obtained from using a particle filter. Moreover, the experiments
demonstrated that OPTIMISTS scales well in high-resolution cases without imposing a significant computational overhead. With the combined advantages of allowing for fast, non-Gaussian, non-linear, high-resolution prediction, the algorithm shows the potential to increase the efficiency of operational prediction systems.

## 1 Introduction

Decision support systems that rely on model-based forecasting of natural phenomena are invaluable to society (Adams et al.,
2003; Penning-Rowsell et al., 2000; Ziervogel et al., 2005). However, despite increasing availability of Earth-sensing data, the problem of estimation or prediction in geophysical systems remains as underdetermined as ever because of the growing complexity of such models (Clark et al., 2017). For example, taking advantage of distributed physics and the mounting availability of computational power, modern models have the potential to more accurately represent impacts of heterogeneities on eco-hydrological processes (Koster et al., 2017). This is achieved through the replacement of lumped representations with

distributed ones, which entails the inclusion of numerous parameters and state variables. The price to pay for thus forsaking parsimony is the added uncertainty in the evaluation of these additional unknowns. Therefore, in order to be able to rely on these high-resolution models for critical real-time and forecast applications, considerable improvements on parameter and initial state estimation techniques must be made with two main goals: First, to allow for an efficient management of the huge

number of unknowns; and second, to mitigate the harmful effects of overfitting—i.e., the loss of forecast skill due to an over-reliance on the calibration/training data (Hawkins, 2004). Because of the numerous degrees of freedom associated with these high-resolution distributed models, overfitting is a much bigger threat due to the phenomenon of equifinality (Beven, 2006).

There exists a plethora of techniques to initialize the state variables of a model through the incorporation of available observations, and they possess overlapping features that make it difficult to develop clear-cut classifications. However, two

main "schools" can be fairly identified: Bayesian data assimilation and variational data assimilation. Bayesian data assimilation creates probabilistic estimates of the state variables in an attempt to also capture their uncertainty. These state probability distributions are adjusted sequentially to better match the observations using Bayes' theorem. While the Kalman filter (KF) is constrained to linear dynamics and Gaussian distributions, ensemble Kalman filters (EnKF) can support non-linear models (Evensen, 2009), and particle filters (PF) can also manage non-Gaussian estimates for added accuracy (Smith et al., 2013).

The stochastic nature of these Bayesian filters is highly valuable because equifinality can rarely be avoided and because of the benefits of quantifying uncertainty in forecasting applications (Verkade and Werner, 2011; Zhu et al., 2002). While superior in accuracy, PFs are usually regarded as impractical for high-dimensional applications (Snyder et al., 2008), and thus recent research has focused on improving their efficiency (van Leeuwen, 2015).

On the other hand, variational data assimilation is more akin to traditional calibration approaches (Efstratiadis and

Koutsoyiannis, 2010) because of its use of optimization methods. It seeks to find a single/deterministic initial state variable combination that minimizes the departures (or "variations") of the modelled values from the observations (Reichle et al., 2001) and, commonly, from their history. One- to three- dimensional variants are also employed sequentially, but the paradigm lends itself easily to evaluating the performance of candidate solutions throughout an extended time window in four-dimensional versions (4D-Var). If the model's dynamics are linearized, the optimum can be very efficiently found in the resulting convex

search space through the use of gradient methods. While this feature has made 4D-Var very popular in meteorology and oceanography (Ghil and Malanotte-Rizzoli, 1991), its application in hydrology has been less widespread because of the difficulty of linearizing land-surface physics (Liu and Gupta, 2007). Moreover, variational data assimilation requires the inclusion of computationally-expensive adjoint models if one wishes to account for the uncertainty of the state estimates (Errico, 1997).

Traditional implementations from both schools have interesting characteristics and thus the development of hybrid methods has received considerable attention (Bannister, 2016). For example, Bayesian filters have been used as adjoints in 4D-Var to enable probabilistic estimates (Zhang et al., 2009). Moreover, some Bayesian approaches have been coupled with optimization techniques to select ensemble members (Dumedah and Coulibaly, 2013; Park et al., 2009). 4DEnVar (Buehner et al., 2010), a fully-hybridized algorithm, is gaining increasing attention for weather prediction (Desroziers et al., 2014; Lorenc et al., 2015).

It is especially interesting that some algorithms have defied the traditional choice between sequential and "extended-time" evaluations. Weak-constrained 4D-Var allows state estimates to be determined at several time steps within the assimilation time window and not only at the beginning (Ning et al., 2014; Trémolet, 2006). Conversely, modifications to EnKFs and PFs have been proposed to extend the analysis of candidate members/particles to span multiple time steps (Evensen and van

Leeuwen, 2000; Noh et al., 2011). The success of these hybrids demonstrates that there is a balance to be sought between the allowed number of degrees of freedom and the amount of information to be assimilated at once.

Following these promising paths, in this article we introduce OPTIMISTS (Optimized PareTo Inverse Modelling through Integrated STochastic Search), a hybrid data assimilation algorithm whose design was guided by the two stated goals: to allow for practical scalability to high-dimensional models, and to enable balancing the imperfect observations and the imperfect

model estimates to minimize overfitting. Table 1 summarizes the main characteristics of typical Bayesian and variational approaches, and their contrasts with those of OPTIMISTS. Our algorithm incorporates the features that the literature has found to be the most valuable from both Bayesian and variational methods while mitigating the deficiencies or disadvantages associated with these original approaches (e.g., the linearity and determinism of 4D-Var and the limited scalability of PFs): Non-Gaussian probabilistic estimation and support for non-linear model dynamics have been long held as advantageous over

their alternatives (Gordon et al., 1993; van Leeuwen, 2009) and, similarly, meteorologists favour extended-period evaluations over sequential ones (Gauthier et al., 2007; Rawlins et al., 2007; Yang et al., 2009). As shown in the table, OPTIMISTS can readily adopt these proven strategies.

However, there are other aspects of the assimilation problem for which no single combination of features has demonstrated its superiority. For example, is the consistency with previous states better achieved through the minimization of a cost function

that includes a background error term (Fisher, 2003), as in variational methods, or through limiting the exploration to samples drawn from that background state distribution, as in Bayesian methods? Table 1 shows that in these cases OPTIMISTS allows for flexible configurations, and it is an additional objective of this study to test which set of feature interactions allows for more accurate forecasts when using highly-distributed models. While many of the concepts utilized within the algorithm have been proposed in the literature before, their combination and broad range of available configurations are unlike those of other

methods, including existing hybrids which have mostly been developed around ensemble Kalman filters and convex optimization techniques (Bannister, 2016)—and therefore limited to Gaussian distributions and linear dynamics.

## 2 Data assimilation algorithm

In this section we describe OPTIMISTS, our proposed data assimilation algorithm which combines advantageous features from several Bayesian and variational methods. As will be explained in detail for each of the steps of the algorithm, these

features were selected with the intent of mitigating the limitations of existing methods. OPTIMISTS allows selecting a flexible data assimilation time step $\Delta t$—i.e., the time window in which candidate state configurations are compared to observations. It can be as short as the model time step, or as long as the entire assimilation window. For each assimilation time step at time $t$

a new state probability distribution $S^{t+\Delta t}$ is estimated from the current distribution $S^t$, the model, and one or more observations $o_{\mathrm{obs}}^{t:t+\Delta t}$. For hydrologic applications, as those explored in this article, these states $S$ include land surface variables within the modelled watershed such as soil moisture, snow cover/water equivalent, and stream water volume; and observations $o$ are typically of streamflow at the outlet (Clark et al., 2008), soil moisture (Houser et al., 1998), and/or snow cover (Andreadis and Lettenmaier, 2006). However, the description of the algorithm will use field-agnostic terminology not to discourage its application in other disciplines.

State probability distributions $S$ in OPTIMISTS are determined from a set of weighted "root" or "base" sample states $s_i$ using multivariate weighted kernel density estimation (West, 1993). This form of non-parametric distributions stands in stark contrast with those from KFs and EnKFs in their ability to model non-Gaussian behaviour—an established advantage of PFs. Each of these samples or ensemble members $s_i$ is comprised of a value vector for the state variables. The objective of the algorithm is then to produce a set of $n$ samples $s_i^{t+\Delta t}$ with corresponding weights $w_i$ for the next assimilation time step to determine the target distribution $S^{t+\Delta t}$.

This process is repeated iteratively each assimilation time step $\Delta t$ until the entire assimilation time frame is covered, at which point the resulting distribution can be used to perform the forecast simulations. In subsection 2.1 we describe the main ideas and steps involved in the OPTIMISTS data assimilation algorithm; details regarding the state probability distributions, mainly on how to generate random samples and evaluate the likelihood of particles, are explained in subsection 2.2; and modifications required for high-dimensional problems are described in subsection 2.3.

**2.1 Description of the OPTIMISTS data assimilation algorithm**

Let a "particle" $P_i$ be defined by a "source" (or initial) vector of state variables $s_i^t$ (which is a sample of distribution $S^t$), a corresponding "target" (or final) state vector $s_i^{t+\Delta t}$ (a sample of distribution $S^{t+\Delta t}$), a set of output values $o_i^{t:t+\Delta t}$ (those that have corresponding observations $o_{\mathrm{obs}}^{t:t+\Delta t}$), a set of fitness metrics $f_i$, a rank $r_i$, and a weight $w_i$. Note that the denomination "particle" stems from the PF literature and is analogous to the "member" term in EnKFs. The fitness metrics $f_i$ are used to compare particles with each other in the light of one or more optimization objectives. The algorithm consists of the following steps, whose motivation and details are included in the subsubsections below and their interactions illustrated in Figure 1. Table 2 lists the meaning of each of the seven global parameters ($\Delta t$, $n$, $w_{\mathrm{root}}$, $p_{\mathrm{samp}}$, $k_{\mathrm{F-class}}$, $n_{\mathrm{evo}}$, and $g$).

1. Drawing: draw root samples $s_i^t$ from $S^t$ in descending weight order until $\sum w_i \geq w_{\mathrm{root}}$
2. Sampling: randomly sample $S^t$ until the total number of samples in the ensemble is $p_{\mathrm{samp}} \times n$
3. Simulation: compute $s_i^{t+\Delta t}$ and $o_i^{t:t+\Delta t}$ from each non-evaluated sample $s_i^t$ using the model
4. Evaluation: compute the fitness values $f_i$ for each particle $P_i$
5. Optimization: create additional samples using evolutionary algorithms and return to 3 (if number of samples is below $n$)
6. Ranking: assign ranks $r_i$ to all particles $P_i$ using non-dominated sorting
7. Weighting: compute the weight $w_i$ for each particle $P_i$ based on its rank $r_i$

### 2.1.1 Drawing step

While traditional PFs draw all the root (or base) samples from $\boldsymbol{S}^t$ (Gordon et al., 1993), OPTIMISTS can limit this selection to a subset of them. The root samples with the highest weight—those that are the "best performers"—are drawn first, then the next ones in descending weight order, until the total weight of the drawn samples $\sum w_i$ reaches $w_{\text{root}}$. $w_{\text{root}}$ thus controls what percentage of the root samples to draw, and, if set to one, all of them are selected.

### 2.1.2 Sampling step

In this step the set of root samples drawn is complemented with random samples. The distinction between root samples and random samples is that the former are those that define the probability distribution $\boldsymbol{S}^t$ (that serve as centroids for the kernels), while the latter are generated stochastically from the kernels. Random samples are generated until the size of the combined set reaches $p_{\text{samp}} \times n$ by following the equations introduced in subsection 2.2. This second step contributes to the diversity of the ensemble in order to avoid sample impoverishment as seen on PFs (Carpenter et al., 1999), and serves as a replacement for traditional resampling strategies (Liu and Chen, 1998). The parameter $w_{\text{root}}$ therefore controls the intensity with which this feature is applied to offer users some level of flexibility. Generating random samples at the beginning, instead of resampling those that have been already evaluated, could lead to discarding degenerate particles (those with high errors) early on and contribute to improved efficiency, given that the ones discarded are mainly those with the lowest weight as determined in the previous assimilation time step.

### 2.1.3 Simulation step

In this step, the algorithm uses the model to compute the resulting state vector $\boldsymbol{s}_i^{t+\Delta t}$ and an additional set of output variables $\boldsymbol{o}_i^{t:t+\Delta t}$ for each of the samples (it is possible that state variables double as output variables). The simulation is executed starting at time $t$ for the duration of the assimilation time step $\Delta t$ (not to be confused with the model time step which is usually shorter). Depending on the complexity of the model, the simulation step can be the one with the highest computational requirements. In those cases, parallelization of the simulations would greatly help in reducing the total footprint of the assimilation process. The construction of each particle $\boldsymbol{P}_i$ is started by assembling the corresponding values computed so far: $\boldsymbol{s}_i^t$ (drawing, sampling, and optimization steps), and $\boldsymbol{s}_i^{t+\Delta t}$ and $\boldsymbol{o}_i^{t:t+\Delta t}$ (simulation step).

### 2.1.4 Evaluation step

In order to determine which initial state $\boldsymbol{s}_i^t$ is the most desirable, a two-term cost function $J$ is typically used in variational methods that simultaneously measures the resulting deviations of modelled values $\boldsymbol{o}_i^{t:t+\Delta t}$ from observed values $\boldsymbol{o}_{\text{obs}}^{t:t+\Delta t}$ and the departures from the background state distribution $\boldsymbol{S}^t$ (Fisher, 2003). The function usually has the form shown in Eq. (1):

$$J_i = c_1 \cdot J_{\text{background}}(\boldsymbol{s}_i^t, \boldsymbol{S}^t) + c_2 \cdot J_{\text{observations}}(\boldsymbol{o}_i^{t:t+\Delta t}, \boldsymbol{o}_{\text{obs}}^{t:t+\Delta t}), \tag{1}$$

where $c_1$ and $c_2$ are balancing constants usually set so that $c_1 = c_2$. Such a multi-criteria evaluation is crucial both to guarantee a good level of fit with the observations (second term) and to avoid the optimization algorithm to produce an initial state that is inconsistent with previous states (first term)—which could potentially result in overfitting problems rooted in disproportionate violations of mass and energy conservation laws (e.g., in hydrologic applications a sharp, unrealistic rise in

the initial soil moisture could reduce $J_{\text{observations}}$ but would increase $J_{\text{background}}$). In Bayesian methods, since the consistency with the history is maintained by sampling only from the prior/background distribution $\boldsymbol{S}^t$, single term functions are used instead—which typically measure the probability density or likelihood of the modelled values given a distribution of the observations.

In OPTIMISTS any such fitness metric could be used and, most importantly, the algorithm allows defining several of them.

Moreover, users can determine whether if each function is to be minimized (e.g., costs or errors) or maximized (e.g., likelihoods). We expect these features to be helpful if one wishes to separate errors when multiple types of observations are available (Montzka et al., 2012) and as a more natural way to consider different fitness criteria (lumping them together in a single function as in Eq. (1) can lead to balancing and "apples and oranges" complications). Moreover, it might prove beneficial to take into account the consistency with the state history both by explicitly defining such an objective here and by allowing

states to be sampled from the previous distribution (and thus compounding the individual mechanisms of Bayesian and variational methods). Functions to measure this consistency are proposed in subsection 2.2. With the set of objective functions defined by the user, the algorithm computes the vector of fitness metrics $\boldsymbol{f}_i$ for each particle during the evaluation step.

### 2.1.5 Optimization step

The optimization step is optional and is used to generate additional particles by exploiting the knowledge encoded in the fitness

values of the current particle ensemble. In a twist to the signature characteristic of variational data assimilation, OPTIMISTS incorporates evolutionary multi-objective optimization algorithms (Deb, 2014) instead of the established gradient-based, single-objective methods. Evolutionary optimizers compensate their slower convergence speed with the capability of efficiently navigating non-convex solution spaces (i.e., the models and the fitness functions do not need to be linear with respect to the observations and the states). This feature effectively opens the door for variational methods to be used in

disciplines where the linearization of the driving dynamics is either impractical, inconvenient, or undesirable. Whereas any traditional multi-objective global optimization method would work, our implementation of OPTIMISTS features a state-of-the-art adaptive ensemble algorithm similar to AMALGAM (Vrugt and Robinson, 2007) that allows model simulations to be run in parallel (Crainic and Toulouse, 2010). The optimizer ensemble includes a genetic algorithm (Deb et al., 2002) and a hybrid approach that combines ant colony optimization (Socha and Dorigo, 2008) and Metropolis-Hastings sampling (Haario

et al., 2001).

During the optimization step, the group of optimizers is used to generate $n_{\text{evo}}$ new sample states $\boldsymbol{s}_i^t$ based on those in the current ensemble. For example, the genetic algorithm selects pairs of base samples with high performance scores $\boldsymbol{f}_i$ and then

proceeds to combine their individual values using standard crossover and mutation operators. The simulation and evaluation steps are repeated for these new samples, and then this iterative process is repeated until the particle ensemble has a size of $n$. Note that $w_{\text{root}}$ and $p_{\text{samp}}$ thus determine what percentage of the particles is generated in which way. For example, for relatively small values of $w_{\text{root}}$ and a $p_{\text{samp}}$ of 0.2, 80% of the particles will be generated by the optimization algorithms. In this way, OPTIMISTS offers its users the flexibility to behave anywhere in the range between "fully Bayesian" ($p_{\text{samp}} = 1$) and "fully variational" ($p_{\text{samp}} = 0$) in terms of particle generation. In the latter case, in which no root and random samples are available, the initial "population"/ensemble of states $s_i^t$ is sampled uniformly from the viable range of each state variable.

### 2.1.6 Ranking step

A fundamental aspect of OPTIMISTS is the way in which it provides a probabilistic interpretation to the results of the multi-objective evaluation, thus bridging the gap between Bayesian and variational assimilation. Such method has been used before (Dumedah et al., 2011) and is based on the employment of non-dominated sorting (Deb, 2014), another technique from the multi-objective optimization literature, which is used to balance the potential tensions between various objectives. This sorting approach is centred on the concept of "dominance," instead of organizing all particles from the "best" to the "worst." A particle dominates another if it outperforms it according to at least one of the criteria/objectives while simultaneously is not outperformed according to any of the others. Following this principle, in the ranking step particles are grouped in "fronts" comprised of members which are mutually non-dominated; that is, none of them is dominated by any of the rest. Particles in a front, therefore, represent the effective trade-offs between the competing criteria.

Figure 1.c illustrates the result of non-dominated sorting applied to nine particles being analysed under two objectives: minimum deviation from observations and maximum likelihood given the background state distribution $S^t$. Note that if a single objective function is used, the sorting method assigns ranks from best to worst according to that function, and two particles would only share ranks if their fitness values coincide. In our implementation we use the fast non-dominated sorting algorithm to define the fronts and assign the corresponding ranks $r_i$ (Deb et al., 2002). More efficient non-dominated sorting alternatives are available if performance becomes an issue (Zhang et al., 2015).

### 2.1.7 Weighting step

In this final step, OPTIMISTS assigns weights $w_i$ to each particle according to its rank $r_i$ as shown in Eqs. (2) and (3). This Gaussian weighting depends on the ensemble size $n$ and the greed parameter $g$, and is similar to the one proposed by Socha and Dorigo (2008). When $g$ is equal to zero, particles in all fronts are weighted uniformly; when $g$ is equal to one, only particles in the Pareto/first front are assigned non-zero weights. With this, the final estimated probability distribution of state variables for the next time step $S^{t+\Delta t}$ can be established using multivariate weighted kernel density estimation (details in the next subsection), as demonstrated in Fig. 1.e., by taking all target states $s_i^{t+\Delta t}$ (circles) as the centroids of the kernels. The

obtained distribution $\boldsymbol{S}^{t+\Delta t}$ can then be used as the initial distribution for a new assimilation time step or, if the end of the assimilation window has been reached, it can be used to perform (ensemble) forecast simulations.

$$w_i = \frac{1}{\sigma\sqrt{2\pi}} e^{-\frac{(r_i-1)^2}{2\sigma^2}} \tag{2}$$

$$\sigma = n \cdot [0.1 + 9.9 \cdot (1-g)^5] \tag{3}$$

## 2.2 Model state probability distributions

As mentioned before, OPTIMISTS uses kernel density probability distributions (West, 1993) to model the stochastic estimates

of the state variable vectors. The algorithm requires two computations related to the state-variable probability distribution $\boldsymbol{S}^t$: obtaining the probability density $p$ or likelihood $\mathcal{L}$ of a sample and generating random samples. The first computation can be used in the evaluation step as an objective function to preserve the consistency of particles with the state history (e.g., to penalize aggressive departures from the prior conditions). It should be noted that several metrics that try to approximate this consistency exist, from very simple (Dumedah et al., 2011) to quite complex (Ning et al., 2014). For example, it is common

in variational data assimilation to utilize the background error term

$$J_{\text{background}} = (\boldsymbol{s} - \boldsymbol{s}_b)^{\text{T}} \mathbf{C}^{-1} (\boldsymbol{s} - \boldsymbol{s}_b), \tag{4}$$

where $\boldsymbol{s}_b$ and $\mathbf{C}$ are the mean and the covariance of the "background" state distribution ($\boldsymbol{S}^t$ in our case) which is assumed to be Gaussian (Fisher, 2003). The term $J_{\text{background}}$ is plugged into the cost function shown in Eq. (1). For OPTIMISTS, we propose that the probability density of the weighted state kernel density distribution $\boldsymbol{S}^t$ at a given point ($p$) be used as a stand-alone objective. The density is given by Eq. (5) (Wand and Jones, 1994). If Gaussian kernels are selected, the kernel function

$K$, parameterized by the bandwidth matrix $\mathbf{B}$, is evaluated using Eq. (6).

$$p(\boldsymbol{s}|\boldsymbol{S}) = \frac{1}{\sum w_i} \sum_{i=1}^{n} [w_i \cdot K_{\mathbf{B}}(\boldsymbol{s} - \boldsymbol{s}_i)] \tag{5}$$

$$K_{\mathbf{B}}^{\text{Gauss}}(\boldsymbol{z}) = \frac{1}{\sqrt{(2\pi)^n \cdot |\mathbf{B}|}} \exp\left(-\frac{1}{2}\boldsymbol{z}^{\text{T}}\mathbf{B}^{-1}\boldsymbol{z}\right) \tag{6}$$

Matrix $\mathbf{B}$ is the covariance matrix of the kernels, and thus determines their spread and orientation in the state space. $\mathbf{B}$ is of size $d \times d$, where $d$ is the dimensionality of the state distribution (i.e., the number of variables), and can be thought of as a scaled-down version of the "background error covariance" matrix $\mathbf{C}$ from the variational literature. In this sense, matrix $\mathbf{B}$, together with the spread of the ensemble of samples $\boldsymbol{s}_i$, effectively encode the uncertainty of the state variables. Several

optimization-based methods exist to compute $\mathbf{B}$ by attempting to minimize the asymptotic mean integrated squared error (AMISE) (Duong and Hazelton, 2005; Sheather and Jones, 1991). However, here we opt to use a simplified approach for the sake of computational efficiency: we determine $\mathbf{B}$ by scaling down the sample covariance matrix $\mathbf{C}$ using Silverman's rule of thumb, which takes into account the number of samples $n$ and the dimensionality of the distribution $d$, as shown in Eq. (7) (Silverman, 1986). Figure 1 shows the density of two two-dimensional example distributions using this method (a and e). If

computational constraints are not a concern, using AMISE-based methods or kernels with variable bandwidth (Hazelton, 2003; Terrell and Scott, 1992) could result in higher accuracy.

$$\mathbf{B}^{\text{Silverman}} = \left(\frac{4}{d+2}\right)^{\frac{2}{d+4}} \cdot n^{-\frac{2}{d+4}} \cdot \mathbf{C} \tag{7}$$

Secondly, OPTIMISTS' sampling step requires generating random samples from a multivariate weighted kernel density distribution. This is achieved by dividing the problem into two: we first select the root sample and then generate a random sample from the kernel associated with that base sample. The first step corresponds to randomly sampling a multinomial distribution with $n$ categories and assigning the normalized weights of the particles as the probability of each category. Once a root sample $\boldsymbol{s}_{\text{root}}$ is selected, a random sample $\boldsymbol{s}_{\text{random}}$ can be generated from a vector $\boldsymbol{v}$ of independent standard normal random values of size $d$ and a matrix $\mathbf{A}$ as shown in Eq. (8). $\mathbf{A}$ can be computed from a Cholesky decomposition (Krishnamoorthy and Menon, 2011) such that $\mathbf{A}\mathbf{A}^{\text{T}} = \mathbf{B}$. Alternatively, an eigendecomposition can be used to obtain $\mathbf{Q}\boldsymbol{\Lambda}\mathbf{Q}^{\text{T}} = \mathbf{B}$ to then set $\mathbf{A} = \mathbf{Q}\boldsymbol{\Lambda}^{\frac{1}{2}}$.

$$\boldsymbol{s}_{\text{random}} = \boldsymbol{s}_{\text{root}} + \mathbf{A}\boldsymbol{v} \tag{8}$$

Both computations (density/likelihood and sampling) require $\mathbf{B}$ to be invertible and, therefore, that none of the variables have zero variance or are perfectly linearly-dependent on each other. Zero-variance variables must therefore be isolated and $\mathbf{B}$ marginalized before attempting to use Eq. (6) or to compute $\mathbf{A}$. Similarly, linear dependencies must also be identified beforehand. If we include variables one by one in the construction of $\mathbf{C}$, we can determine if a newly added one is linearly dependent if the determinant of the extended sample covariance matrix $\mathbf{C}$ is zero. Once identified, the regression coefficients for the dependent variable can be efficiently computed from $\mathbf{C}$ following the method described by Friedman et al. (2008). The constant coefficient of the regression must also be calculated for future reference. What this process effectively does is to determine a linear model for each dependent variable that is represented by a set of regression coefficients. Dependent variables are not included in $\mathbf{C}$, but they need to be taken into account afterwards (e.g., by determining their values for the random samples by solving the linear model with the values obtained for the variables in $\mathbf{C}$).

### 2.3 High-dimensional state vectors

When the state vector of the model becomes large (i.e., $d$ increases), as is the case for distributed high-resolution numerical models, difficulties start to arise when dealing with the computations involving the probability distribution. At first, the probability density, as computed with Eqs. (5) and (6), tends to diverge either towards zero or towards infinity. This phenomenon is related to the normalization of the density—so that it can integrate to one—and to its fast exponential decay as a function of the sample's distance from the kernel's centres. In these cases we propose replacing the density computation with an approximated likelihood formulation that is proportional to the inverse square Mahalanobis distance (Mahalanobis, 1936) to the root samples, thus skipping the exponentiation and normalization operations of the Gaussian density. This simplification, which corresponds to the inverse square difference between the sample value and the kernel's mean in the univariate case, is

shown in Eq. (9). The resulting distortion of the Gaussian bell-curve shape does not affect the results significantly, given that OPTIMISTS uses the fitness functions only to check for domination between particles—so only the sign of the differences between likelihood values are important and not their actual magnitudes.

$$\mathcal{L}^{\text{Mahalanobis}}(\boldsymbol{s}|\boldsymbol{S}) = \frac{1}{\sum w_i} \sum_{i=1}^{n} \frac{w_i}{|(\boldsymbol{s} - \boldsymbol{s}_i)^{\mathrm{T}} \mathbf{B}^{-1} (\boldsymbol{s} - \boldsymbol{s}_i)|} \tag{9}$$

However, computational constraints might also make this simplified approach unfeasible both due to the $O(d^2)$ space requirements for storing the bandwidth matrix $\mathbf{B}$ and the $O(d^3)$ time complexity of the decomposition algorithms, which rapidly become huge burdens for the memory and the processors. Therefore, we can chose to sacrifice some accuracy by using a diagonal bandwidth matrix $\mathbf{B}$ which does not include any covariance term—only the variance terms in the diagonal are computed and stored. This implies that, even though the multiplicity of root samples would help in maintaining a large portion of the covariance, another portion is lost by preventing the kernels from reflecting the existing correlations. In other words, variables would not be rendered completely independent, but rather conditionally independent because the kernels are still centred on the set of root samples. Kernels using diagonal bandwidth matrices are referred to as "D-class" while those using the full covariance matrix are referred to as "F-class." The $k_{\text{F-class}}$ parameter controls which version is used.

With only the diagonal terms of matrix $\mathbf{B}$ available ($b_{jj}$), we opt to roughly approximate the likelihood by computing the average of the standardized marginal likelihood value for each variable $j$, as shown in Eq. (10):

$$\mathcal{L}^{\text{independent}}(\boldsymbol{s}|\boldsymbol{S}) = \frac{1}{d\sqrt{2\pi}\sum w_i} \sum_{j=1}^{d} \sum_{i=1}^{n} \left\{ w_i \cdot \exp\left[ -\frac{(s_j - s_{i,j})^2}{2b_{jj}} \right] \right\}, \tag{10}$$

where $s_j$ represents the $j^{\text{th}}$ element of state vector $\boldsymbol{s}$, and $s_{i,j}$ represents the $j^{\text{th}}$ element of the $i^{\text{th}}$ sample of probability distribution $\boldsymbol{S}$. Independent/marginal random sampling of each variable can also be applied to replace Eq. (8) by adding random Gaussian residuals to the elements of the selected root sample $\boldsymbol{s}_{\text{root}}$. Sparse bandwidth matrices (Friedman et al., 2008; Ghil and Malanotte-Rizzoli, 1991) or low-rank approximations (Bannister, 2008; Ghorbanidehno et al., 2015; Li et al., 2015) could be worthwhile intermediate alternatives to our proposed quasi-independent approach to be explored in the future.

## 3 Experimental setup

In this section we prepare the elements to investigate whether if OPTIMISTS can help improve the forecasting skill of hydrologic models. More specifically, the experiments seek to answer the following questions: Which characteristics of Bayesian and variational methods are the most advantageous? How can OPTIMISTS be configured to take advantage of these characteristics? How does the algorithm compare to established data assimilation methods? And how does it perform with high-dimensional applications? To help answer these questions, this section first introduces two case studies and then it describes a traditional PF that was used for comparison purposes.

**3.1 Case studies**

We coupled a Java implementation of OPTIMISTS with two popular open-source distributed hydrologic modelling engines: Variable Infiltration Capacity (VIC) (Liang et al., 1994, 1996b, 1996a, Liang and Xie, 2001, 2003) and the Distributed Hydrology Soil and Vegetation Model (DHSVM) (Wigmosta et al., 1994, 2002). VIC is targeted at large watersheds by focusing on vertical subsurface dynamics, and also enabling intra-cell precipitation, soil, and vegetation heterogeneity. The DHSVM, on the other hand, was conceived for high-resolution representations of the Earth's surface, allowing for saturated and unsaturated subsurface flow routing and 1D/2D surface routing (Zhang et al., 2018). Both engines needed several modifications so that they could be executed in a non-continuous fashion as required for sequential assimilation. Given the non-Markovian nature of surface routing schemes coupled with VIC that are based either on multiscale approaches (Guo et al., 2004; Wen et al., 2012) or on the unit hydrograph concept (Lohmann et al., 1998), a simplified routing routine was developed that treats the model cells as channels—albeit with longer retention times. In the simplified method, direct runoff and baseflow produced by each model cell is partly routed through an assumed "equivalent" channel (slow component) and partly poured directly to the channel network (fast component). Both the channel network and the equivalent channels representing overland flow hydraulics are modelled using the Muskingum method. On the other hand, several important bugs in version 3.2.1 of the DHSVM, mostly related to the initialization of state variables but also pertaining to routing data and physics, were fixed.

We selected two watersheds to perform streamflow forecasting tests using OPTIMISTS: one with the VIC model running at a 1/8$^{th}$ degree resolution for the Blue River in Oklahoma, and the other with the DHSVM running at a 100 m resolution for the Indiantown Run in Pennsylvania. Table 3 lists the main characteristics of the two test watersheds and the information of their associated model configurations. Figure 2 shows the land cover map together with the layout of the modelling cells for the two watersheds. The multi-objective ensemble optimization algorithm included in OPTIMISTS was employed to calibrate the parameters of the two models with the streamflow measurements from the corresponding USGS stations. For the Blue River, the traditional $\ell_2$-norm Nash-Sutcliffe Efficiency ($NSE_{\ell_2}$) (which focuses mostly on the peaks of hydrographs), an $\ell_1$-norm version of the Nash-Sutcliffe Efficiency coefficient ($NSE_{\ell_1}$) (Krause et al., 2005), and the mean absolute relative error MARE (which focuses mostly on the inter-peak periods) were used as optimization criteria. From 85,600 candidate parameterizations tried, one was chosen from the resulting Pareto front with $NSE_{\ell_2}$ = 0.69, $NSE_{\ell_1}$ = 0.56, and MARE = 44.71%. For the Indiantown Run, the $NSE_{\ell_2}$, MARE, and absolute bias were optimized, resulting in a parameterization, out of 2,575, with $NSE_{\ell_2}$ = 0.81, MARE = 37.85%, and an absolute bias of 11.83 l/s.

These "optimal" parameter sets, together with additional sets produced in the optimization process were used to run the models and determine a set of time-lagged state variable vectors $s$ to construct the state probability distribution $S^0$ at the beginning of each of a set of data assimilation scenarios. The state variables include liquid and solid interception; ponding, water equivalent and temperature of the snow packs; and moisture and temperature of each of the soil layers. While we do not expect all of these variables to be identifiable and sensitive within the assimilation problem, we decided to be thorough in their inclusion—

a decision that also increases the challenge for the algorithm in terms of the potential for overfitting. The Blue River model application has 20 cells, with a maximum of seven intra-cell soil/vegetation partitions. After adding the stream network variables, the model has a total of $d = 812$ state variables. The Indiantown Run model application has a total of 1,472 cells and $d = 33,455$ state variables.

Three diverse scenarios were selected for the Blue River, each of them comprised of a two-week assimilation period (when streamflow observations are assimilated), and a two-week forecasting period (when the model is run in an open loop using the states obtained at the end of the assimilation period): Scenario 1, starting on October 15[th], 1996, is rainy through the entire four weeks. Scenario 2, which starts on January 15[th], 1997, has a dry assimilation period and a mildly rainy forecast period. Scenario 3, starting on June 1[st], 1997, has a relatively rainy assimilation period and a mostly-dry forecast period. Two scenarios, also

spanning four weeks, were selected for the Indiantown Run, one starting on July 26[th], 2009 and the other on August 26[th], 2009. We used factorial experiments (Montgomery, 2012) to test different configurations of OPTIMISTS on each of these scenarios, by first assimilating the streamflow and then measuring the forecasting skill. In this type of experimental designs a set of assignments is established for each parameter and then all possible assignment combinations are tried. The design allows to establish the statistical significance of altering several parameters simultaneously, providing an adequate framework for

determining, for example, whether if using a short or a long assimilation time step $\Delta t$ is preferable, or if utilizing the optional optimization step within the algorithm is worthwhile. Table 4 shows the setup of each of the three full factorial experiments we conducted, together with the selected set of assignments for OPTIMISTS' parameters. The forecasts were produced in an ensemble fashion, by running the models using each of the samples $s_i$ from the state distribution $S$ at the end of the assimilation time period, and then using the samples' weights $w_i$ to produce an average forecast. Deterministic model parameters (those

from the calibrated models) and forcings were used in all simulations.

Observation errors are usually taken into account in traditional assimilation algorithms by assuming a probability distribution for the observations at each time step, and then performing a probabilistic evaluation of the predicted value of each particle/member against that distribution. As mentioned in section 2 such a fitness metric, like the likelihood utilized in PFs to weight candidate particles, is perfectly compatible with OPTIMISTS. However, since it is difficult to estimate the magnitude

of the observation error in general, and fitness metrics $f_i$ here are only used to determine (non-)dominance between particles, we opted to use the mean absolute error (MAE) with respect to the streamflow observations in all cases.

For the Blue River scenarios, a secondary likelihood objective/metric was used in some cases to select for particles with higher consistency with the history. It was computed using either Eq. (10) if $k_{F-class}$ was set to false, or Eq. (9) if it was set to true. Equation (10) was used for all Indiantown Run scenarios given the large number of dimensions. The assimilation period was

of two weeks for most configurations, except for those in Experiment 3 which have $\Delta t = 4$ weeks. During both the assimilation and the forecasting periods we used unaltered streamflow data from the USGS and forcing data from NLDAS-2 (Cosgrove et al., 2003)—even though a forecasted forcing would be used instead in an operational setting (e.g., from systems like NAM (Rogers et al., 2009) or ECMWF (Molteni et al., 1996)). While adopting perfect forcings for the forecast period leads to an overestimation of their accuracy, any comparisons with control runs or between methods are still valid as they all share the

same benefit. Also, removing the uncertainty in the meteorological forcings allows the analysis to focus on the uncertainty corresponding to the land surface.

## 3.2 Data assimilation method comparison

Comparing the performance of different configurations of OPTIMISTS can shed light into the adequacy of individual strategies utilized by traditional Bayesian and variational methods. For example, producing all particles with the optimization algorithms ($p_{samp} = 0$), setting long values for $\Delta t$, and utilizing a traditional two-term cost function as that in Eq. (1), makes the method behave somewhat as a hard-constrained 4D-Var approach; while sampling all particles from the source state distribution ($p_{samp}$ = 1), setting $\Delta t$ equal to the model time step, and using a single likelihood objective involving the observation error, would resemble a PF. Herein we also compare OPTIMISTS with a traditional PF on both model applications. Since the forcing is assumed to be deterministic, the implemented PF uses Gaussian "regularization"/perturbation of resampled particles to avoid degeneration (Pham, 2001). Resampling is executed such that the probability of duplicating a particle is proportional to their weight (Moradkhani et al., 2012).

Additionally, the comparison is performed using a continuous forecasting experiment setup instead of a scenario-based one. In this continuous test, forecasts are performed every time step and compiled in series for different forecast lead times that span several months. Forecast lead times are of 1, 3, 6, and 12 days for the Blue River and of 6 hours, and 1, 4, and 16 days for the Indiantown Run. Before each forecast, both OPTIMISTS and the PF assimilate streamflow observations for the assimilation time step of each algorithm (daily for the PF). The assimilation is performed cumulatively, meaning that the initial state distribution $\boldsymbol{S}^t$ was produced by assimilating all the records available since the beginning of the experiment until time $t$. The forecasted streamflow series are then compared to the actual measurements to evaluate their quality using deterministic metrics ($NSE_{\ell_2}$, $NSE_{\ell_1}$, and MARE) and two probabilistic ones: the ensemble-based continuous ranked probability score (CRPS) (Bröcker, 2012), which is computed for each time step and then averaged for the entire duration of the forecast; and the average normalized probability density $p$ of the observed streamflow $q_{obs}$ given the distribution of the forecasted ensemble $\boldsymbol{q}_{forecast}$:

$$p(q_{obs}|\boldsymbol{q}_{forecast}) = \frac{\sum_{i=1}^{n} w_i \cdot (2\pi b^2)^{-2} \cdot \exp[-(q_{obs} - q_i)^2/(2b^2)]}{\sum_{i=1}^{n} w_i}, \tag{11}$$

where the forecasted streamflow $\boldsymbol{q}_{forecast}$ is composed of values $q_i$ for each particle $i$ and accompanying weight $w_i$, and $b$ is the bandwidth of the univariate kernel density estimate. $b$ can be obtained utilizing Silverman's rule of thumb (Silverman, 1986). The probability $p$ is computed every time step, then normalized by multiplying by the standard deviation of the estimate, and then averaged for all time steps. As opposed to the CRPS, which can only give an idea of the bias of the estimate, the density $p$ can detect both bias and under/over-confidence: high values for the density indicate that the ensemble is producing narrow estimates around the true value, while low values indicate either that the stochastic estimate is spread too thin or is centred far away from the true value.

## 4 Results and discussion

This section summarizes the forecasting results obtained from the three scenario-based experiments and the continuous forecasting experiments on the Blue River and the Indiantown Run model applications. The scenario-based experiments were performed to explore the effects of multiple parameterizations of OPTIMISTS, and the performance was analysed as follows.

5 The model was run for the duration of the forecast period (two weeks) using the state configuration encoded in each root state $s_i$ of the distribution $S$ obtained at the end of the assimilation period for each configuration of OPTIMISTS and each scenario. We then computed the mean streamflow time series for each case by averaging the model results for each particle $P_i$ (the average was weighted based on the corresponding weights $w_i$). With this averaged streamflow series, we compute the three performance metrics—the $\text{NSE}_{\ell_2}$, the $\text{NSE}_{\ell_1}$, and the MARE—based on the observations from the corresponding stream gauge.

10 The values for each experiment, scenario, and configuration are listed in tables in the supplementary material. With these, we compute the change in the forecast performance between each configuration and a control open-loop model run (one without the benefit of assimilating the observations).

### 4.1 Blue River – low resolution application

The supplementary material includes the performance metrics for all the tested configurations on all scenarios and for all 15 scenario-based experiments. Figure 3 summarizes the results for Experiment 1 with the VIC model application for the Blue River watershed, in which the distributions of the changes in MARE after marginalizing the results for each scenario and each of the parameter assignments are shown. That is, each box (and pair of whiskers) represents the distribution of change in MARE of all cases in the specified scenario or for which the specified parameter assignment was used. Negative values in the vertical axis indicate that OPTIMISTS decreased the error, while positive values indicate it increased the error. It can be seen that, on 20 average, OPTIMISTS improves the precision of the forecast in most cases, except for several of the configurations in Scenario 1 (for this scenario the control already produces a good forecast) and when using an assimilation step $\Delta t$ of one day. We performed an analysis of variance (ANOVA) to determine the statistical significance of the difference found for each of the factors indicated in the horizontal axis. While Figure 3 shows the $p$-values for the main effects, the full ANOVA table for all experiments can be found in the supplementary material. From the values in Figure 3, we can conclude that the assimilation 25 time step, the number of objectives, and the use of optimization algorithms are all statistically significant. On the other hand, the number of particles and the use of F-class kernels are not.

A $\Delta t$ of five days produced the best results overall for the tested case, suggesting that there exists a sweet spot that balances the amount of information being assimilated (larger for a long $\Delta t$), and the number of state variables to be modified (larger for a small $\Delta t$). Based on such results, it is reasonable to assume that the sweet spot may depend on the time series of precipitation, 30 the characteristics of the watershed, and the temporal and spatial resolutions of its model application. From this perspective, the poor results for a step of one day could be explained in terms of overfitting, where there are many degrees of freedom and only one value being assimilated per step. Evaluating particles in the light of two objectives, one minimizing departures from

the observations and the other maximizing the likelihood of the source state, resulted in statistically-significant improvements compared to using the first objective alone. Additionally, the data suggests that not executing the optional optimization step of the algorithm ("optimization = false"), but instead relying only on particles sampled from the prior/source distribution, is also beneficial. These two results reinforce the idea that maintaining consistency with the history to some extent is of paramount importance, perhaps to the point where the strategies used in Bayesian filters and variational methods are insufficient in isolation. Indeed, the best performance was observed only when both sampling was limited to generate particles from the prior state distribution and the particles were evaluated for their consistency with that distribution.

On the other hand, we found it counterintuitive that neither using a larger particle ensemble nor taking into account state-variable dependencies through the use of F-class kernels lead to improved results. In the first case it could be hypothesized that using too many particles could lead to overfitting, since there would be more chances of particles being generated that happen to match the observations better but for the "wrong reasons." In the second case, the non-parametric nature of kernel density estimation could be sufficient for encoding the raw dependencies between variables, especially in low-resolution cases like this one, in which significant correlations between variables in adjacent cells are not expected to be too high. Both results deserve further investigation, especially concerning the impact of D- vs. F-class kernels in high-dimensional models.

Interestingly, the ANOVA also yielded small $p$-values for several high-order interactions (see the ANOVA table in the supplementary material). This means that, unlike the general case for factorial experiments as characterized by the sparsity-of-effects principle (Montgomery et al., 2009), specific combinations of multiple parameters have a large effect on the forecasting skill of the model. Significant interactions (with $p$ smaller than 0.05) are between the objectives and $\Delta t$ ($p = 0.001$); $n$ and $k_{\text{F-class}}$ ($p = 0.039$); $\Delta t$ and the use of optimization ($p = 0.000$); the use of optimization and $k_{\text{F-class}}$ ($p = 0.029$); the objectives, $\Delta t$, and the use of optimization ($p = 0.043$); $n$, $\Delta t$, and $k_{\text{F-class}}$ ($p = 0.020$); $n$, the use of optimization, and $k_{\text{F-class}}$ ($p = 0.013$); and $n$, $\Delta t$, the use of optimizers, and $k_{\text{F-class}}$ ($p = 0.006$). These interactions show that, for example, using a single objective is especially inadequate when the time step is of one day or when optimization is used. Also, employing optimization is only significantly detrimental when $\Delta t$ is of one day—probably because of intensified overfitting, and that choosing F-class kernels leads to higher errors when $\Delta t$ is small, $n$ large, and the optimizers are being used.

Based on these results, we recommend the use of both objectives and no optimization as the preferred configuration of OPTIMISTS for the Blue River application. A time step of around five days appears to be adequate for this specific model application. Also, without strong evidence for their advantages, we recommend using more particles or kernels of class F only if there is no pressure for computational frugality. However, the number of particles should not be too small to ensure an appropriate sample size.

Table 5 shows the results of the five-months-long continuous forecasting experiment on the Blue River using a 30-particle PF and a configuration of OPTIMISTS with a 7-day assimilation time step $\Delta t$, three objectives (NSE$_{\ell_2}$, MARE, and the likelihood), 30 particles, no optimization, and D-class kernels. This specific configuration of OPTIMISTS was chosen from a few that were tested with the recommendations above applied. The selected configuration was the one that best balanced the spread

and the accuracy of the ensemble as some configurations had slightly better deterministic performance but larger ensemble spread for dry weather—which lead to worse probabilistic performance.

Figure 4 shows the probabilistic streamflow forecasts for both algorithms for a lead time of 6 days. The portrayed evolution of the density, in which the mean does not necessarily correspond to the centre of the ensemble spread, evidences the non-Gaussian nature of both estimates. Both the selected configuration of OPTIMISTS and the PF methods show relatively good performance for all lead times (1, 3, 6, and 12 days) based on the performance metrics. However, the PF generally outperforms OPTIMISTS. We offer three possible explanations for this result. First, the relatively low dimensionality of this test case does not allow OPTIMISTS to showcase its real strength, perhaps especially since the large scale of the watershed does not allow for tight spatial interactions between state variables. Second, OPTIMISTS can find solutions based on multiple objectives rather than a single one, which could be advantageous when multiple types of observations are available (e.g., of streamflow, evapotranspiration, and soil moisture). Thus, the solutions are likely not the best for each individual objective, but the algorithm balances their overall behaviour across the multiple objectives. Due to the lack of observations on multiple variables, only streamflow observations are used in these experiments even though more than one objective is used. Since it is the case that these objectives are consistent with each other, to a large extent, for the studied watershed, the strengths of using multiple objectives within the Pareto approach in OPTIMISTS cannot be fully evidenced. Third, additional efforts might be needed to find a configuration of the algorithm, together with a set of objectives, that best suit the specific conditions of the tested watershed. While PFs remain easier to use "out of the box" because of their ease of configuration, the fact that adjusting the parameters of OPTIMISTS allowed to trade-off deterministic and probabilistic accuracy points to the adaptability potential of the algorithm. This allows for probing the spectrum between exploration and exploitation of candidate particles—which usually leads to higher and lower diversity of the ensemble, respectively.

## 4.2 Indiantown Run – high resolution application

Figure 5 summarizes the changes in performance when using OPTIMISTS in Experiment 2. In this case, the more uniform forcing and streamflow conditions of the two scenarios allowed to statistically analyse all three performance metrics. For Scenario 1 we can see that OPTIMISTS produces a general increase in the Nash-Sutcliffe coefficients, but a decline in the MARE, evidencing tension between fitting the peaks and the inter-peak periods simultaneously. For both scenarios there are configurations that performed very poorly, and we can look at the marginalized results in the boxplots for clues into which parameters might have caused this. Similar to the Blue River case, the use of a 1-hour time step significantly reduced the forecast skill, while the longer step almost always improved it; and the inclusion of the secondary history-consistency objective ("2 objectives") also resulted in improved performance. Not only does it seem that for this watershed the secondary objective mitigated the effects of overfitting, but it was interesting to note some configurations in which using it actually helped to achieve a better fit during the assimilation period.

While the ANOVA also provided evidence against the use of optimization algorithms, we are reluctant to instantly rule them out on the grounds that there were statistically significant interactions with other parameters (see the ANOVA table in the

supplementary material). The optimizers led to poor results in cases with one-hour time steps or when only the first objective was used. Other statistically significant results point to the benefits of using the root samples more intensively (in opposition to using random samples) and, to a lesser extent, to the benefits of maintaining an ensemble of moderate size.

Figure 6 shows the summarized changes in Experiment 3, where the effect of the time step $\Delta t$ is explored in greater detail. Once again, there appears to be evidence favouring the hypothesis that there exists a sweet spot, and in this case it appears to be close to the two weeks mark: both shorter and longer time steps led to considerably poorer performance. In this experiment, with all configurations using both optimization objectives, we can see that there are no clear disadvantages of using optimization algorithms (but also no advantages). Experiment 3 also shows that the effect of the greed parameter $g$ is not very significant. That is, selecting some particles from dominated fronts to construct the target state distribution, and not only from the Pareto front, does not seem to affect the results.

Table 6 and Figure 7 show the results from comparing continuous forecasts from the PF and from a configuration of OPTIMISTS with a time step of one week, two objectives, 50 particles, and no optimization. Both algorithms display overconfidence in their estimations, which is evidenced in Figure 7 by the bias and narrowness of the ensembles' spread. It is possible that a more realistic incorporation of uncertainties pertaining to model parameters and forcings (which, as mentioned, are trivialized in these tests) would help compensate overconfidence. For the time being, these experiments help characterize the performance of OPTIMISTS in contrast with the PF, as both algorithms are deployed under the same circumstances. In this sense, while the forecasts obtained using the PF show slightly better results for lead times of 6 hours and 1 day, OPTIMISTS shows a better characterization of the ensemble's uncertainty for the longer lead times.

OPTIMISTS' improved results in the high-resolution test case over those in the low-resolution one suggest that the strengths of the hybrid method might become more apparent as the dimensionality, and therefore the difficulty, of the assimilation problem increases. However, while OPTIMISTS was able to produce comparable results to those of the PF, it was not able to provide definite advantages in terms of accuracy. As suggested before, additional efforts might be needed to find the configurations of OPTIMISTS that better match the characteristics of the individual case studies. Moreover, the implemented version of the PF did not present the particle degeneracy or impoverishment problems usually associated with these filters when dealing with high dimensionality, which also prompts further investigation.

## 4.3 Computational performance

It is worth noting that the longer the assimilation time step, the faster the entire process is. This occurs because, even though the number of hydrological calculations is the same in the end, for every assimilation time step the model files need to be generated accordingly, then accessed, and finally the result files written and accessed. This whole process takes a considerable amount of time. Therefore, everything else being constant, sequential assimilation (like with PFs) automatically imposes additional computational requirements. In our tests we used RAM drive software to accelerate the process of running the models sequentially and, even then, the overhead imposed by OPTIMISTS was consistently below 10% of the total computation time. Most of the computational effort remained with running the model, both for VIC and the DHSVM. In this

sense, model developers may consider allowing their engines to be able to receive input data from main memory, if possible, to facilitate data assimilation and other similar processes.

## 4.4 Recommendations for configuring OPTIMISTS

Finally, here we summarize the recommended choices for the parameters in OPTIMISTS based on the results of the experiments. In the first place, given their low observed effect, default values can be used for $g$ (around 0.5). A $w_{\text{root}}$ higher than 90% was found to be advantageous. The execution of the optimization step ($p_{\text{samp}} < 1$) was, on the other hand, not found to be advantageous and, therefore, we consider it a cleaner approach to simply generate all samples from the initial distribution. Similarly, while not found to be disadvantageous, using diagonal bandwidth (D-class) kernels provide a significant improvement in computational efficiency and are thus recommended for the time being. Future work will be conducted to further explore the effect of the bandwidth configuration in OPTIMISTS.

Even though only two objective functions were tested, one measuring the departures from the observations being assimilated and another measuring the compatibility of initial samples with the initial distribution, the results clearly show that it is beneficial to simultaneously evaluate candidate particles using both criteria. While traditional cost functions like the one in Eq. (1) do indeed consider both aspects, we argue that that using multiple objectives has the added benefit of enriching the diversity of the particle ensemble and, ultimately, the resulting probabilistic estimate of the target states.

Our results demonstrated that the assimilation time step is the most sensitive parameter and, therefore, its selection must be done with the greatest involvement. Taken the results together, we recommend that multiple choices be tried for any new case study looking to strike a balance between the amount of information being assimilated and the number of degrees of freedom. This empirical selection should also be performed with a rough sense of what is the range of forecasting lead-times that is considered the most important. Lastly, more work is required to provide guidelines to select the number of particles $n$ to be used. While the literature suggests that more should increase forecast accuracy, our tests did not back this conclusion. We tentatively recommend trying different ensemble sizes based on the computational resources available and selecting the one that offers the best observed trade-off between accuracy and efficiency.

## 5 Conclusions and future work

In this article we introduced OPTIMISTS, a flexible, model-independent data assimilation algorithm that effectively combines the signature elements from Bayesian and variational methods: By employing essential features from particle filters, it allows performing probabilistic non-Gaussian estimates of state variables through the filtering of a set of particles drawn from a prior distribution to better match the available observations. Adding critical features from variational methods, OPTIMISTS grants its users the option of exploring the state space using optimization techniques and evaluating candidate states through a time window of arbitrary length. The algorithm fuses a multi-objective/Pareto analysis of candidate particles with kernel density probability distributions to effectively bridge the gap between the probabilistic and the variational perspectives. Moreover, the

use of evolutionary optimization algorithms enables its efficient application on highly non-linear models as those usually found in most geosciences. This unique combination of features represent a clear differentiation from the existing hybrid assimilation methods in the literature (Bannister, 2016), which are limited to Gaussian distributions and linear dynamics.

We conducted a set of hydrologic forecasting factorial experiments on two watersheds, the Blue River with 812 state variables and the Indiantown Run with 33,455, at two distinct modelling resolutions using two different modelling engines: VIC and the DHSVM, respectively. Capitalizing on the flexible configurations available for OPTIMISTS, these tests allowed to determine which individual characteristics of traditional algorithms prove to be the most advantageous for forecasting applications. For example, while there is a general consensus in the literature favouring extended time steps (4D) over sequential ones (1D-3D), the results from assimilating streamflow data in our experiments suggest that there is an ideal duration of the assimilation time step that is dependent on the case study under consideration, on the spatiotemporal resolution of the corresponding model application, and on the desired forecast length. Sequential time steps not only required considerably longer computational times but also produced the worst results—perhaps given the overwhelming number of degrees of freedom in contrast with the scarce observations available. Similarly, there was a drop in the performance of the forecast ensemble when the algorithm was set to use overly long time steps.

Procuring the consistency of candidate particles, not only with the observations but also with the history, led to significant gains in predictive skill. OPTIMISTS can be configured to both perform Bayesian sampling and find Pareto-optimal particles that trade-off deviations from the observations and from the prior conditions. This Bayesian/multi-objective formulation of the optimization problem was especially beneficial for the high-resolution watershed application, as it allows the model to overcome the risk of overfitting generated by the enlarged effect of equifinality.

On the other hand, our experiments did not produce enough evidence to recommend neither exploring the state space with optimization algorithms instead of doing so with simple probabilistic sampling, the use of a larger number of particles above the established baseline of 100, nor the computationally-intensive utilization of full covariance matrices to encode the dependencies between variables in the kernel-based state distributions. Nevertheless, strong interactions between several of these parameters suggest that some specific combinations could potentially yield strong outcomes. Together with OPTIMISTS' observed high level of sensitivity to the parameters, these results indicate that there could be promise in the implementation of self-adaptive strategies (Karafotias et al., 2014) to assist in their selection in the future. With these experiments, we were able to configure the algorithm to consistently improve the forecasting skill of the models compared to control open-loop runs. Additionally, comparative tests showed that OPTIMISTS was able to reliably produce adequate forecasts that were comparable to those resulting from assimilating the observations with a particle filter in the high-resolution application. While not being able to provide consistent accuracy advantages over the implemented particle filter, OPTIMISTS does offer considerable gains in computational efficiency given its ability to analyse multiple model time steps each time.

Moreover, in this article we offered several alternatives in the implementation of the components of OPTIMISTS whenever there were tensions between prediction accuracy and computational efficiency. In the future, we will focus on incorporating additional successful ideas from diverse assimilation algorithms and on improving components in such a way that both of these

goals are attained with ever-smaller compromises. For instance, the estimation of initial states should not be overburdened with the responsibility of compensating structural and calibration deficiencies in the model. In this sense, we embrace the vision of a unified framework for the joint probabilistic estimation of structures, parameters, and state variables (Liu and Gupta, 2007), where it is important to address challenges associated with approaches that would increase the indeterminacy of the

problem by adding unknowns without providing additional information or additional means of relating existing variables. We expect that with continued efforts OPTIMISTS will be a worthy candidate framework to be deployed in operational settings for hydrologic prediction and beyond.

**Data and code availability**

All the data utilized to construct the models is publicly available through the internet from their corresponding US government

agencies' websites. The Java implementation of OPTIMISTS and of the particle filter are available through GitHub (https://github.com/felherc/). These sources include all the information needed to replicate the experiments in this article.

**Acknowledgements**

The authors are thankful to the two anonymous referees and the Editor for their valuable comments and suggestions. This work was supported in part by the United States Department of Transportation through award #OASRTRS-14-H-PIT to the

University of Pittsburgh and by the William Kepler Whiteford Professorship from the University of Pittsburgh.

**Previous versions**

An earlier version of this article was submitted for peer review to be considered for publication on HESS and is available in the HESSD archive (Hernández and Liang, 2016). While the proposed method itself has seen no changes since, this new version attempts to make its presentation much more approachable and has included the comparative tests with the particle

filter.

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

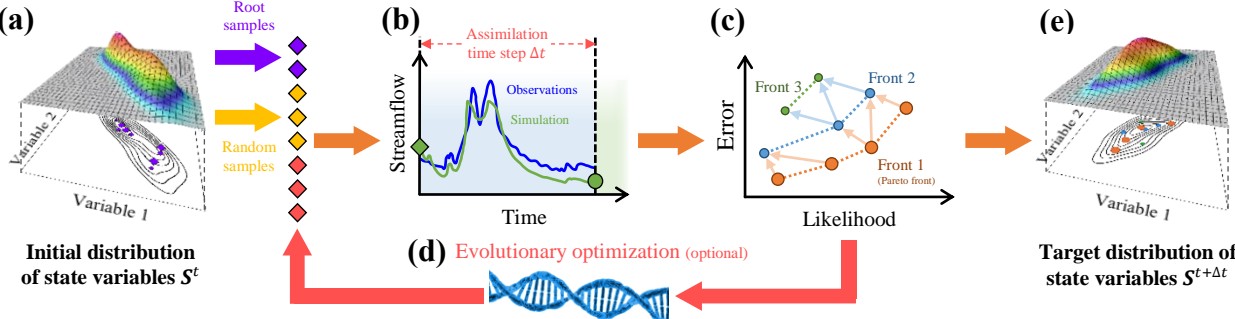

**Figure 1. Steps in OPTIMISTS, to be repeated for each assimilation time step $\Delta t$.** In this example state vectors have two variables, observations are of streamflow, and particles are judged using two user-selected objectives: the likelihood given $S^t$ to be maximized and the error given the observations to be minimized. **(a)** Initial state kernel density distribution $S^t$ from which root samples (purple rhombi) are taken during the drawing step and random samples (yellow rhombi) are taken during the sampling step. **(b)** Execution of the model (simulation step) for each source sample for a time equal to $\Delta t$ to compute output variables (for comparison with observations) and target samples (circles). **(c)** Evaluation of each particle (evaluation step) based on the objectives and organization into non-domination fronts (ranking step). The dashed lines represent the fronts while the arrows denote domination relationships between particles in adjacent fronts. **(d)** Optional optimization step which can be executed several times and that uses a population-based evolutionary optimization algorithm to generate additional samples (red rhombi). **(e)** Target state kernel density distribution $S^{t+\Delta t}$ constructed from the particles' final samples (circles) after being weighted according to the rank of their front (weighting step): kernels centred on samples with higher weight (shown larger) have a higher probability density contribution.

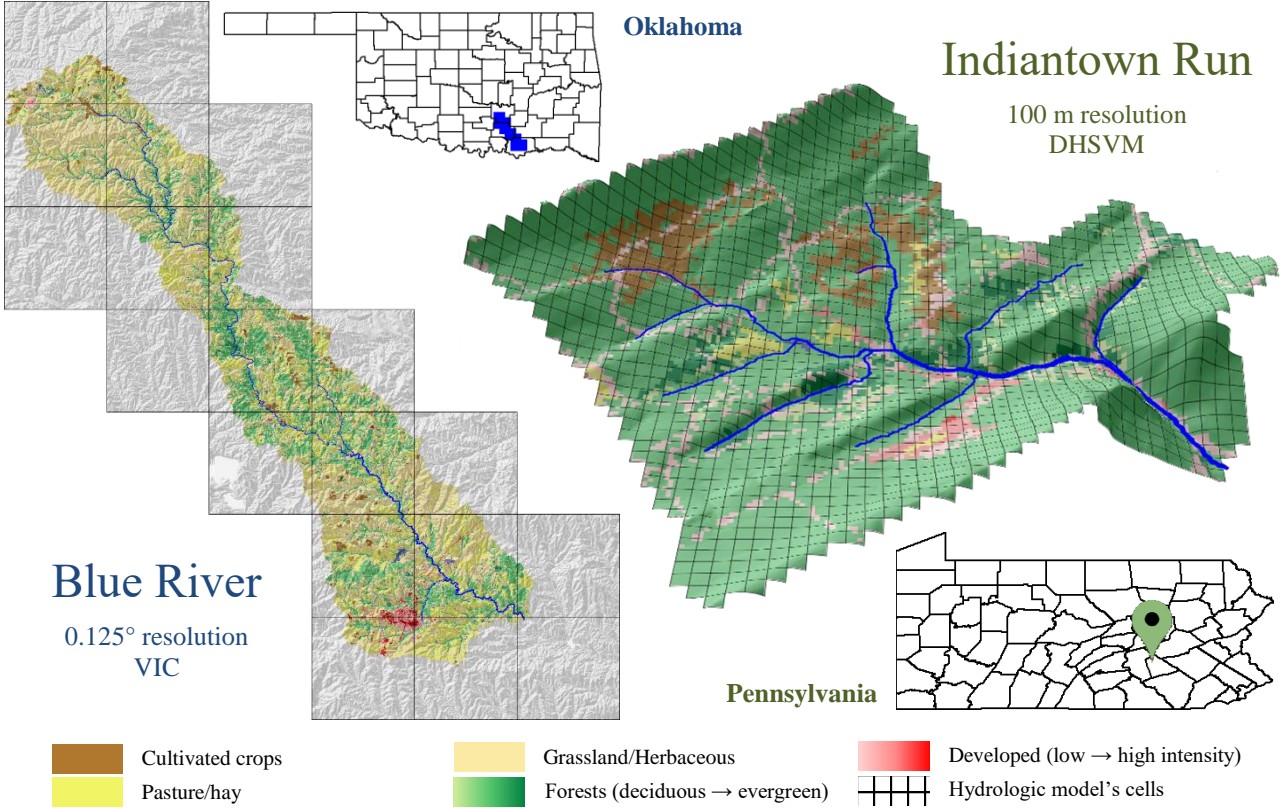

**Figure 2. Maps of the two test watersheds in the United States displaying the 30 m resolution land cover distribution from the NLCD (Homer et al., 2012). Left: Oklahoma's Blue River watershed 0.125° resolution VIC model application (20 cells). Right: Pennsylvania's Indiantown Run watershed 100 m resolution DHSVM model application (1,472 cells).**

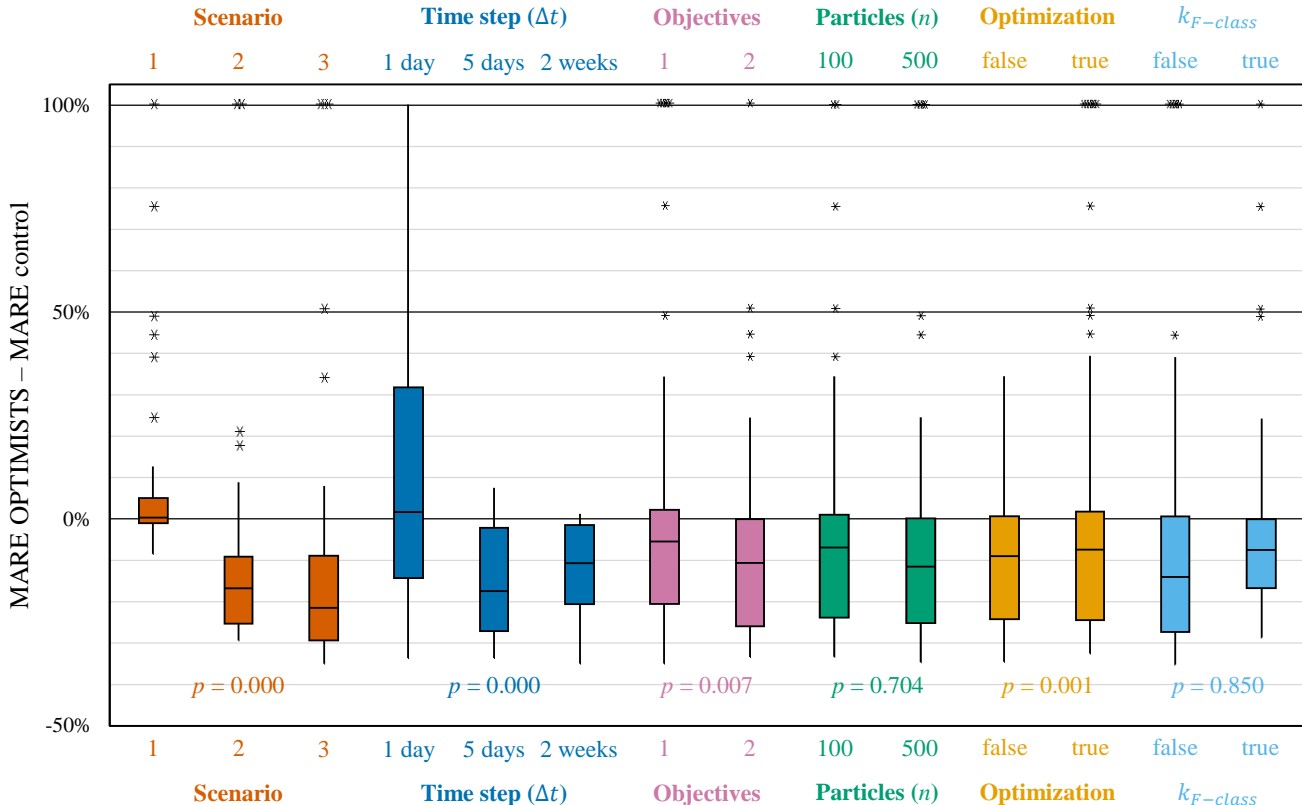

**Figure 3. Boxplots of the changes in forecasting error (MARE) achieved while using OPTIMISTS on Experiment 1 (Blue River). Changes are relative to an open-loop control run where no assimilation was performed. Each column corresponds to the distribution of the error changes on the specified scenario or assignment to the indicated parameter. Positive values indicate that OPTIMISTS increased the error, while negative values indicate it decreased the error. Outliers are noted as asterisks and values were limited to 100%. For the one-objective case the particles' MAE was to be minimized; for the two-objective case, the likelihood given the background was to be maximized in addition. No optimization ("false") corresponds to $p_{samp} = 1.0$ (i.e., all samples are obtained from the prior distribution); "true" corresponds to $p_{samp} = 0.25$. The $p$-values were determined using ANOVA (Montgomery, 2012), and indicate the probability that the differences in means corresponding to boxes of the same colour are produced by chance (e.g., values close to zero indicate certainty that the parameter effectively affects the forecast error).**

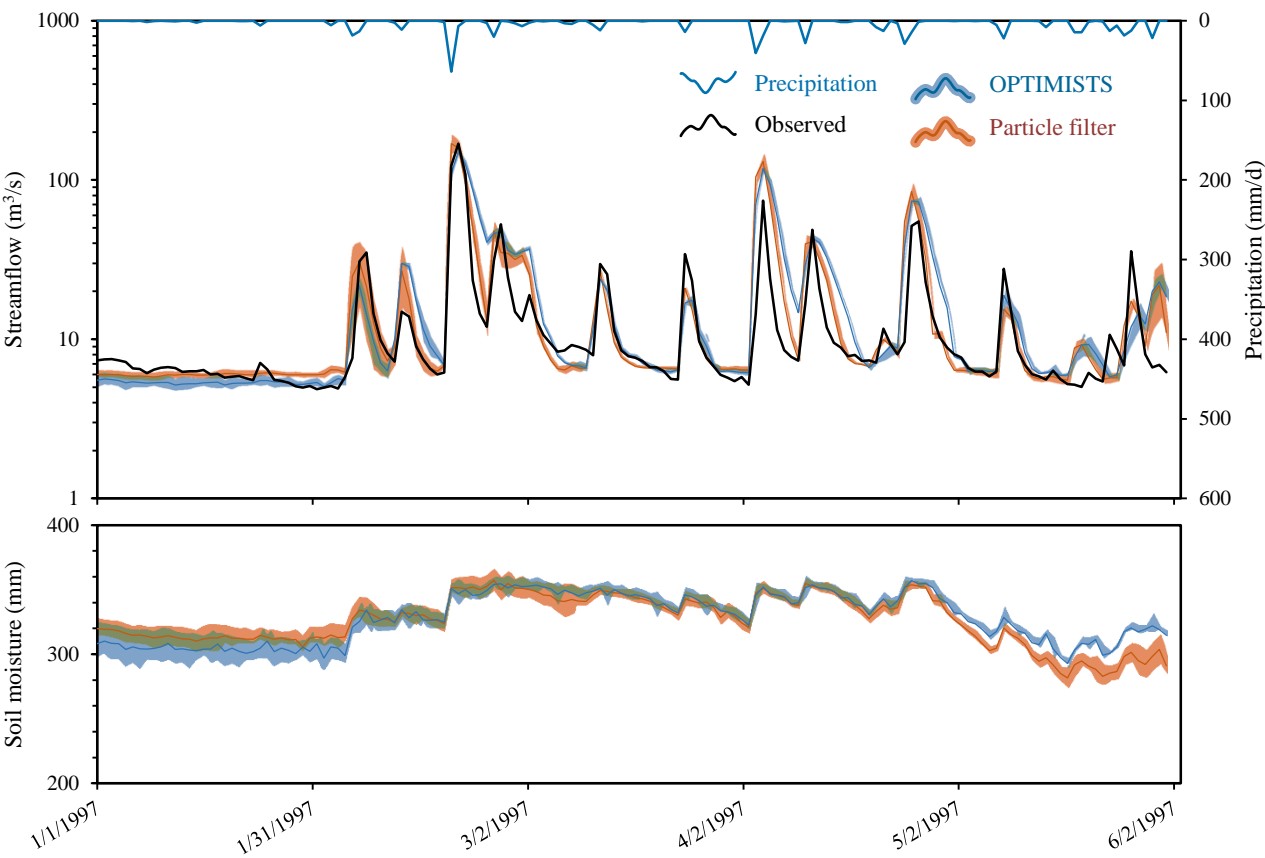

**Figure 4. Comparison of six-day lead time probabilistic streamflow (top) and area-averaged soil moisture (bottom) forecasts between OPTIMISTS** ($\Delta t$ = 7 days; 3 objectives: NSE$_{\ell 2}$, MARE, and likelihood; $n$ = 30; no optimization, and D-class kernels) **and a traditional PF** ($n$ = 30) **for the Blue River. The dark blue and orange lines indicate the mean of OPTIMISTS' and the PF's ensembles respectively, while the light blue and light orange bands illustrate the spread of the forecast by highlighting the areas where the probability density of the estimate is at least 50% of the density at the mode (the maximum) at that time step. The green bands indicate areas where the light blue and light orange bands intersect.**

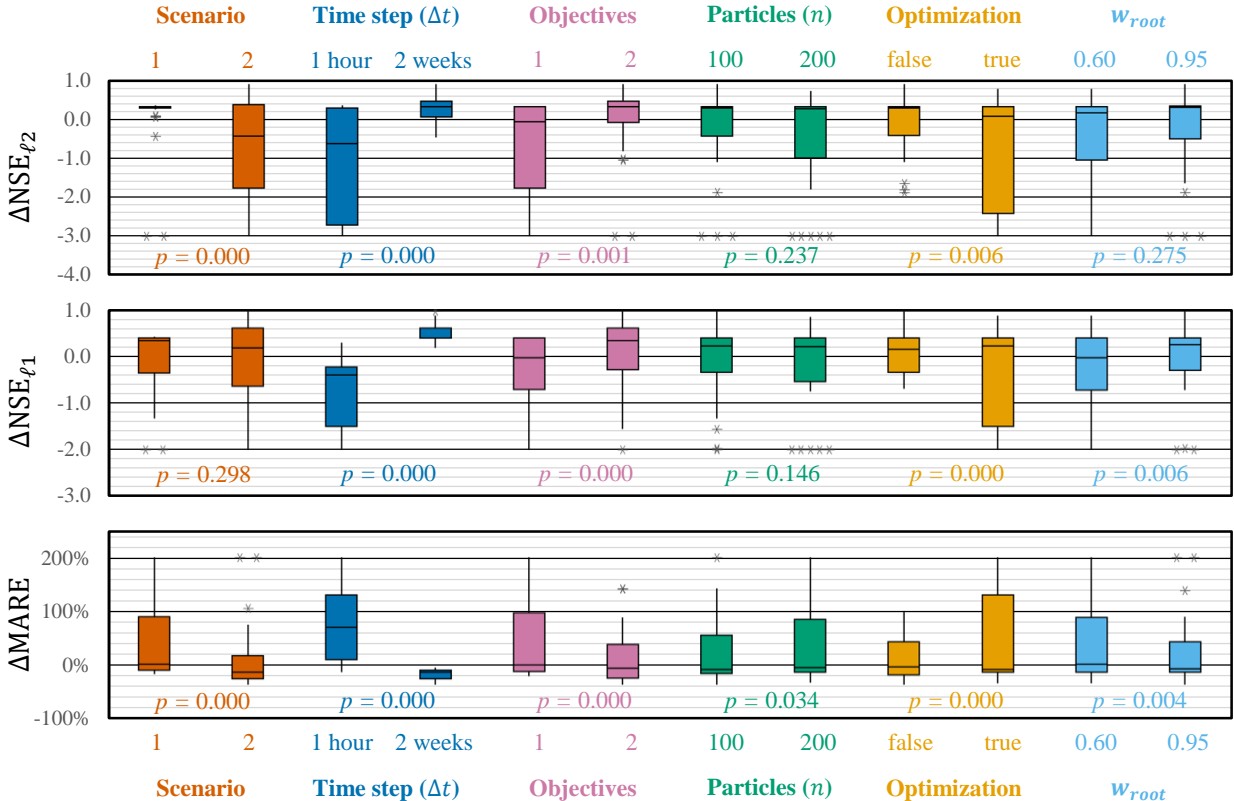

**Figure 5. Boxplots of the changes in forecasting performance (NSE$_{\ell2}$, NSE$_{\ell1}$, and MARE) achieved while using OPTIMISTS on Experiment 2 (Indiantown Run). Changes are relative to an open-loop control run where no assimilation was performed. Each column corresponds to the distribution of the error metric changes on the specified scenario or assignment to the indicated parameter. Outliers are noted as stars and values were constrained to NSE$_{\ell2} \geq -3$, NSE$_{\ell1} \geq -3$, and MARE $\leq 200\%$. Positive values indicate improvements for the NSE$_{\ell2}$ and the NSE$_{\ell1}$. The meaning for the MARE and for other symbols are the same as those defined in Fig. 3.**

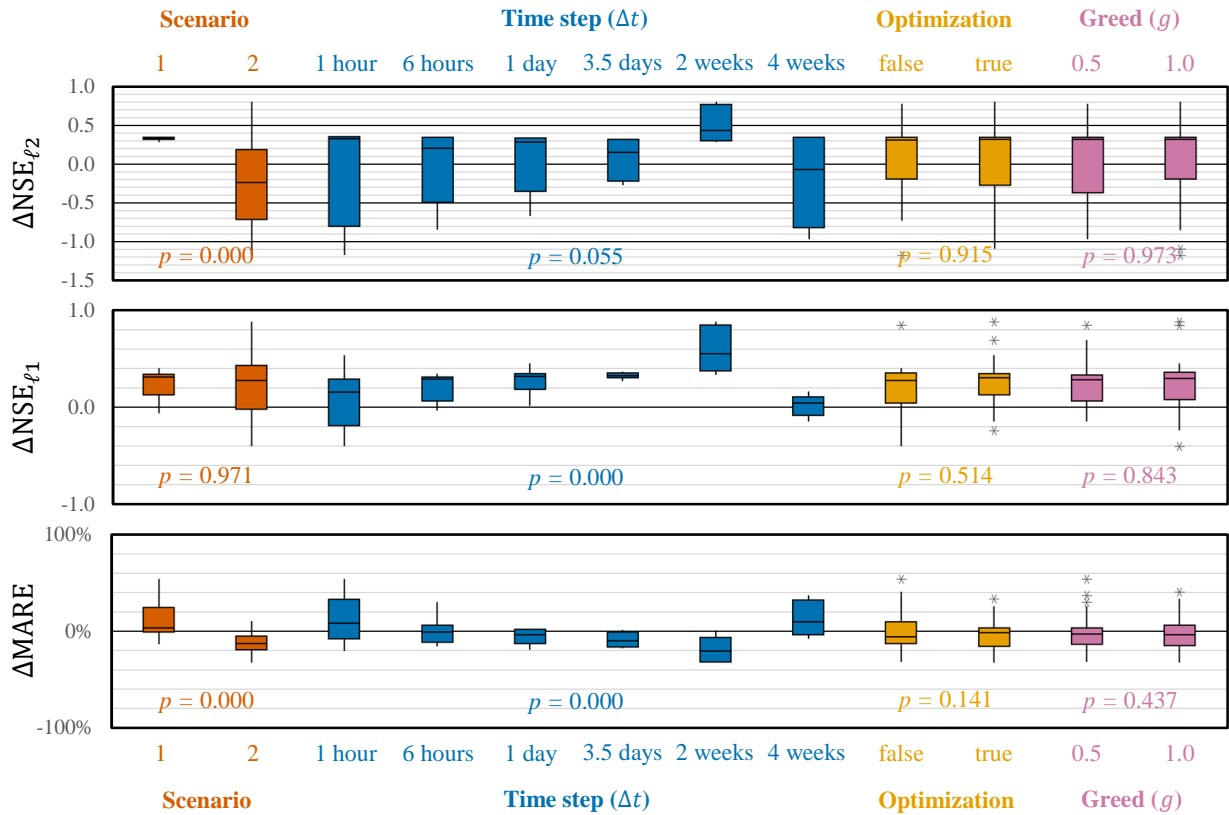

**Figure 6. Boxplots of the changes in forecasting performance ($NSE_{\ell 2}$, $NSE_{\ell 1}$, and MARE) achieved while using OPTIMISTS on Experiment 3 (Indiantown Run). Changes are relative to an open-loop control run where no assimilation was performed. Each column corresponds to the distribution of the error metric changes on the specified scenario or assignment to the indicated parameter. Positive values indicate improvements for the $NSE_{\ell 2}$ and the $NSE_{\ell 1}$. See the caption of Fig. 3 for more information.**

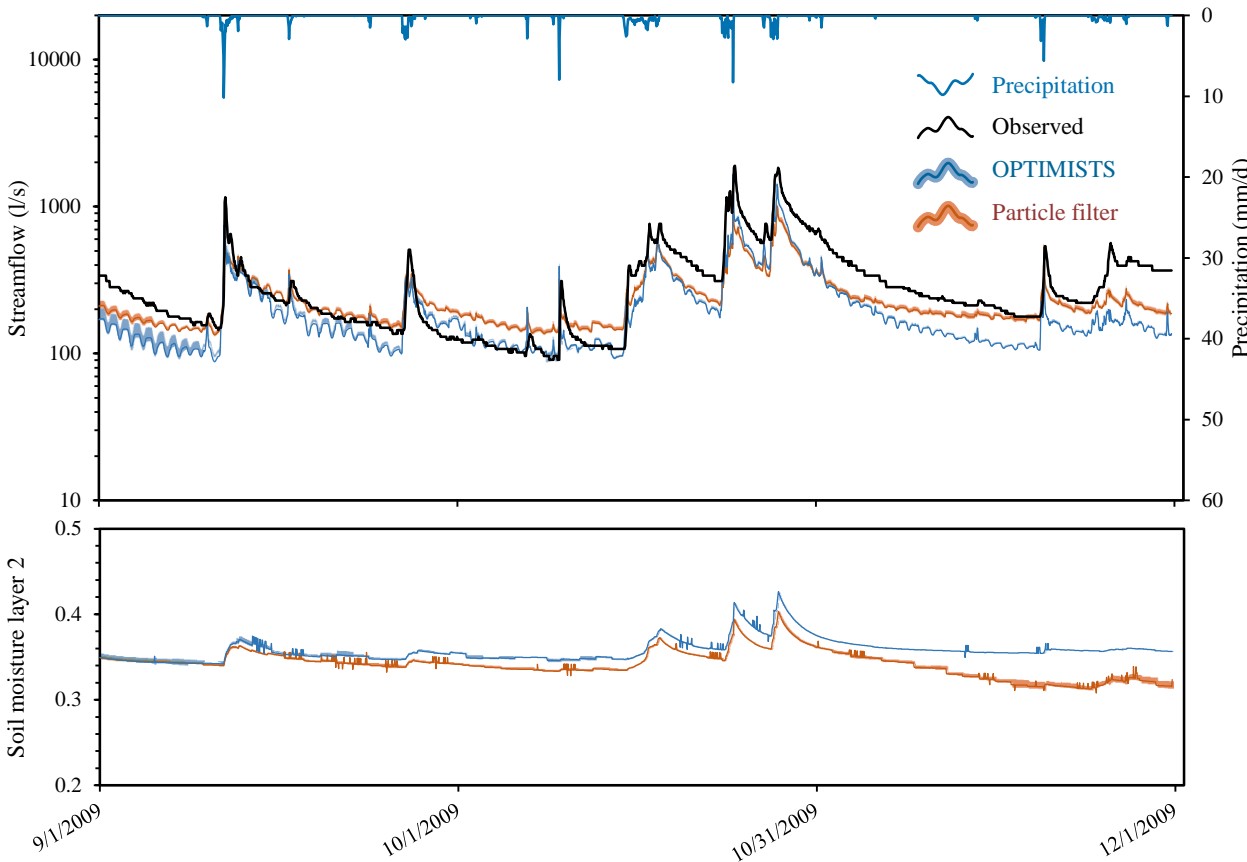

**Figure 7. Comparison of four-day lead time probabilistic streamflow (top) and area-averaged soil moisture (bottom) forecasts between OPTIMISTS ($\Delta t$ = 7 days, 2 objectives, $n$ = 50, no optimization, and D-class kernels) and a traditional PF ($n$ = 50) for the Indiantown Run. The dark blue and orange lines indicate the mean of OPTIMISTS' and the PF's ensembles respectively, while the light blue and light orange bands illustrate the spread of the forecast by highlighting the areas where the probability density of the estimate is at least 50% of the density at the mode (the maximum) at that time step. The green bands indicate areas where the light blue and light orange bands intersect. Layer 2 of the soil corresponds to 100 to 250 mm depths.**

**Table 1. Comparison between the main features of standard Bayesian data assimilation algorithms (KF: Kalman Filter, EnKF: Ensemble KF, PF: Particle Filter), variational data assimilation (one- to four-dimensional), and OPTIMISTS.**

|  | **Bayesian** | **Variational** | **OPTIMISTS** |
|---|---|---|---|
| **Resulting state-variable estimate** | Probabilistic: Gaussian (KF, EnKF), Non-Gaussian (PF) | Deterministic (unless adjoint model is used) | Probabilistic (using kernel density estimation) |
| **Solution quality criteria** | High likelihood given observations | Minimum cost value (error, departure from history) | Flexible: e.g., min. error, max. consistency with history |
| **Analysis time step** | Sequential | Sequential (1D-3D) or entire assimilation window (4D) | Flexible |
| **Search method** | Iterative Bayesian belief propagation | Convex optimization | Coupled belief propagation/multi-objective optimization |
| **Model dynamics** | Linear (KF), non-linear (EnKF, PF) | Linearized to obtain convex solution space | Non-linear (non-convex solution space) |

**Table 2. List of global parameters in OPTIMISTS**

| Symbol | Description | Range |
|---|---|---|
| $\Delta t$ | Assimilation time step (particle evaluation time frame) | $\mathbb{R}^+$ |
| $n$ | Total number of root states $\boldsymbol{s}_i$ in the probability distributions | $\mathbb{N} \geq 2$ |
| $w_{\text{root}}$ | Total weight of root samples drawn from $\boldsymbol{S}^t$ | $\mathbb{R} \in [0, 1]$ |
| $p_{\text{samp}}$ | Percentage of $n$ corresponding to drawn and random samples | $\mathbb{R} \in [0, 1]$ |
| $k_{\text{F-class}}$ | Whether or not to use F-class kernels. If not: D-class kernels. | true or false |
| $n_{\text{evo}}$ | Samples to be generated by the optimizers per iteration | $\mathbb{N} \geq 2$ |
| $g$ | Level of greed for the assignment of particle weights $w_i$ | $\mathbb{R} \in [-1, 1]$ |

**Table 3. Characteristics of the two test watersheds: Blue River and Indiantown Run. US hydrologic units are defined in Seaber et al. (1987). Elevation information was obtained from the Shuttle Radar Topography Mission (Rodríguez et al., 2006); land cover and impervious percentage from the National Land Cover Database (Homer et al., 2012); soil type from CONUS-SOIL (Miller and White, 1998); and precipitation, evapotranspiration, and temperature from NLDAS-2 (Cosgrove et al., 2003). The streamflow and temperature include their range of variation of 90% of the time (5% tails at the high and low end are excluded).**

| Model characteristic | Blue River | Indiantown Run |
|---|---|---|
| USGS station; US hydrologic unit | 07332500; 11140102 | 01572950; 02050305 |
| Area (km$^2$); impervious | 3,031; 8.05% | 14.78; 0.83% |
| Elevation range; average slope | 158 m – 403 m; 3.5% | 153 m – 412 m; 14.5% |
| Land cover | 43% grassland, 28% forest, 21% pasture/hay | 74.6% deciduous forest |
| Soil type | Clay loam (26.4%), clay (24.8%), sandy loam (20.26%) | Silt loam (51%), sandy loam (49%) |
| Avg. streamflow (90% range) | 9.06 m$^3$/s (0.59 m$^3$/s – 44.71 m$^3$/s) | 0.3 m$^3$/s (0.035 m$^3$/s – 0.793 m$^3$/s) |
| Avg. precipitation; avg. ET | 1,086 mm/year; 748 mm/year | 1,176 mm/year; 528 mm/year |
| Avg. temperature (90% range) | 17.26°C (2.5°C – 31°C) | 10.9°C (-3.5°C – 24°C) |
| Model cells; stream segments; $d$ | 20; 14; 812 | 1,472; 21; 33,455 |
| Resolution | 0.125°; daily | 100 m; hourly |
| Calibration | 167 parameters; 85 months; objectives: NSE$_{\ell_2}$, NSE$_{\ell_1}$, MARE | 18 parameters; 20 months; objectives: NSE$_{\ell_2}$, MARE, absolute bias |

**Table 4. Setup of the three factorial experiments, including the watershed, the total number of configurations (conf.), the values assigned to OPTIMISTS' parameters, and which objectives (objs.) were used (one objective: minimize MAE given the streamflow observations; two objectives: minimize MAE and maximize likelihood given source/background state distribution $S^t$). $n_{\text{evo}}$ was set to 25 in all cases. The total number of configurations results from combining all the possible parameter assignments listed for each experiment. Note that for Experiment 3 there are configurations that require a four-week assimilation period (all others have a length of two weeks).**

| No. | Watershed | Conf. | $\Delta t$ | $n$ | $w_{\text{root}}$ | $p_{\text{samp}}$ | $k_{\text{F-class}}$ | $g$ | objs. |
|---|---|---|---|---|---|---|---|---|---|
| 1 | Blue River | 48 | 1d, 5d, 2w | 100, 500 | 0.95 | 0.25, 1 | false, true | 0.75 | 1, 2 |
| 2 | Indiantown Run | 32 | 1h, 2w | 100, 200 | 0.6, 0.95 | 0.25, 1 | false | 0.75 | 1, 2 |
| 3 | Indiantown Run | 24 | 1h, 6h, 1d, 3.5d, 2w, 4w | 100 | 0.95 | 0.4, 1 | false | 0.5, 1 | 2 |

**Table 5.** Continuous daily streamflow forecast performance metrics for the Blue River application using OPTIMISTS ($\Delta t$ = 7 days, 3 objectives: $NSE_{\ell_2}$, MARE, and likelihood; $n$ = 30; no optimization; and D-class kernels) and a traditional PF ($n$ = 30). The continuous forecast extends from January to June, 1997. The $NSE_{\ell_2}$, $NSE_{\ell_1}$, and MARE (deterministic) are computed using the mean streamflow of the forecast ensembles and contrasting it with the daily observations, while the CRPS and the density (probabilistic) are computed taking into account all the members of the forecasted ensemble.

| Algorithm | Lead time | $NSE_{\ell 2}$ | $NSE_{\ell 1}$ | MARE | CRPS ($m^3$/s) | Density |
|---|---|---|---|---|---|---|
| OPTIMISTS | 1 day | 0.497 | 0.293 | 51.40% | 7.173 | 0.061 |
| | 3 days | 0.527 | 0.312 | 50.16% | 6.959 | 0.065 |
| | 6 days | 0.534 | 0.315 | 50.18% | 6.945 | 0.073 |
| | 12 days | 0.516 | 0.297 | 51.26% | 7.124 | 0.078 |
| Particle filter | 1 day | 0.675 | 0.522 | 30.06% | 4.480 | 0.098 |
| | 3 days | 0.623 | 0.493 | 33.20% | 4.744 | 0.113 |
| | 6 days | 0.602 | 0.473 | 35.79% | 5.000 | 0.109 |
| | 12 days | 0.515 | 0.432 | 38.36% | 5.593 | 0.105 |

**Table 6.** Continuous hourly streamflow forecast performance metrics for the Indiantown Run application using OPTIMISTS ($\Delta t$ = 7 days, 2 objectives; $n$ = 50; no optimization; and D-class kernels) and a traditional PF ($n$ = 50). The continuous forecast extends from September to December, 2009. The $NSE_{\ell_2}$, $NSE_{\ell_1}$, and MARE (deterministic) are computed using the mean streamflow of the forecast ensembles and contrasting it with the daily observations, while the CRPS and the density (probabilistic) are computed taking into account all the members of the forecasted ensemble.

| Algorithm | Lead time | $NSE_{\ell 2}$ | $NSE_{\ell 1}$ | MARE | CRPS (l/s) | Density |
|---|---|---|---|---|---|---|
| OPTIMISTS | 6 hours | 0.574 | 0.316 | 32.25% | 97.27 | 0.016 |
| | 1 day | 0.609 | 0.340 | 31.42% | 93.92 | 0.013 |
| | 4 days | 0.573 | 0.316 | 32.20% | 97.19 | 0.025 |
| | 16 days | 0.521 | 0.272 | 33.90% | 103.51 | 0.013 |
| Particle filter | 6 hours | 0.660 | 0.480 | 26.87% | 79.61 | 0.061 |
| | 1 day | 0.639 | 0.464 | 26.68% | 82.75 | 0.051 |
| | 4 days | 0.558 | 0.401 | 27.42% | 93.20 | 0.021 |
| | 16 days | 0.520 | 0.346 | 28.75% | 102.37 | 0.010 |