# Peer review of "Hybridizing Bayesian and variational data assimilation for highresolution hydrologic forecasting"

_Hydrology and Earth System Sciences, 2017_

## Referee Comment (RC1) · Anonymous Referee #1 · 13 Oct 2017

I really enjoyed reading the paper, which deals with the important issue of improving model updating techniques for better flood predictions. This manuscript proposes a new data assimilation procedure which combines Bayesian and variational approaches. I believe this is an important contribute to DA research field and of notable interest and modernity, especially for the HESS readership. However, I still have some comments regarding method, structure, and readability of the paper. Below you can find some major comments:

- Results of this research are not well described in the abstract and it is somehow difficult to grasp the main advantage of this approach.

[Figure]

- From the introduction, it is not really clear the difference between OPTIMIST and other hybrid 4D approaches. Novelty has to be better explained in order to further appreciate the added value of such method.

- Nowadays, there are many DA methods with varying complexities and accuracies. However, few of these methods are used in early warning system to improve flood predictions. Did the authors investigate the way to easily implement OPTIMISTS by water authorities for flood forecasting in any existing early warning system? Is there any advantage in terms of computational time if compared to PF and 4D-Var?

- Page 1, lines 14-16: Is this sentence related to the watershed's location or to the use of different models for different case studies? Authors should clarify this point.

- Page 3, lines 8-9: The authors mentioned that "a hybrid data assimilation algorithm that incorporates the most valuable features from both Bayesian and variational methods ". Which ones are these valuable features?

- Description of Table 1 should be better included within the paper. Right now it looks quite disconnected from the other part of the introduction.

- I found very difficult to follow the flow of thoughts of the authors in describing the DA method. I think it will be beneficial for the readability of the paper to include in section 2 a figure representing the structure of OPTIMISTS. In addition, authors tend to use complex terms for non-DA expert. I suggest revising the description of the paper in order to make it "accessible" to everyone and increase its impact on the scientific community.

- At this point, results are valid only for the 5 considered flood events and 2 basins obtained. As expected, results largely depend on the features of the flood events and quality of rainfall data. I am afraid that the samll number of events makes results rather random. I suggest to increase the number of flood events to make more general conclusions for this study.
- A crucial component in each DA application is the proper definition of model and observational error. While model error is accurately described, I could not see a clear definition of the observations error (standard deviation in Eq.10). The authors have to include more information and references about it.

- Are you using actual meteorological forecasts or are you using the observations as perfect forecasts? Please specify

- I suggest the authors to split results and discussions in two different sections, this would make reading the text so much easier.

---

## Referee Comment (RC2) · Anonymous Referee #2 · 21 Dec 2017

Before beginning review of this manuscript, although not mentioned in the text and reference, this should be considered a re-submission of the previous HESSD manuscript entitled "Hybridizing sequential and variational data assimilation for robust high-resolution hydrologic forecasting (https://doi.org/10.5194/hess-2016-454)" by the same authors, which was rejected in 2016. I suggest the editorial board compare the final revision of the previous HESSD manuscript with the current one if the track record was not screened yet. I cannot examine whether the authors submitted the final revision in the previous submission in 2016 or not. However, if so, improvement and uniqueness of the current manuscript over the rejected final manuscript should be carefully evaluated. In addition, since HESSD is independent publication, the previous manuscript in

2016 should be cited and discussed in this manuscript.

This manuscript proposed a hybrid DA method, OPTIMISTS, combining sequential and variational methods, and compared performance of developed methodology over PF and VAR using distributed hydrologic models. The topic is of interest to a wide range of hydrologic modelling community. The strategy of the proposed methodology to leverage different DA approaches, sequential and variational DA, is one of the important trends in recent studies. However, there are major gaps in experimental setup and evaluation, and incomplete reasoning in new methodology which require significant changes before publication. I hope the followings would be helpful to improve the quality of manuscript.

1) Evaluation period and methods In this manuscript, the total evaluation period is 10 weeks (5 cases with a 2-weeks period each): 3 scenarios for 2-weeks forecasts in the Blue River and 2 scenarios for 2-weeks forecasts in the Indiantown Run, not including assimilation period.

The evaluation period for hydrologic modelling and data assimilation is usually longer than at least 6-8 months and up to multiple decades. The total 10-weeks forecasts (2-weeks piecewise each) and associated metrics cannot be accepted as a rigorous evaluation.

Given that the selected events in the Blue River in the 2016 manuscript are different from those in the current one, there seems to a potential to further increase evaluation period. In Table 3, the authors also mentioned calibration periods are 85 and 20 months, respectively.

Considering the availability of observation data, what is the maximum evaluation period for two catchments? Why don't you use the whole or most calibration period for DA evaluation? Was there any reason to use the limited period for evaluation?

For the larger domain, the Blue River catchment, is there just one streamflow observa-

tion gage over 3,000 square kilometer area? Why don't you assimilate observations in multiple locations to reduce equifinality and overfitting?

In this study, evaluation metrics were estimated for the whole 2-weeks forecast period. However, it is more common to evaluate metrics for varying forecast lead times because the impact of updating varies and disappears over time.

I highly suggest the evaluation period and method should be reconsidered to qualify a kind of general standard shown in many forecast and DA-related papers: simulating more than several months for each catchment and evaluating metrics for varying forecast lead times.

2) Probabilistic evaluation Although the proposed method is a stochastic approach, probabilistic metrics were not measured and analyzed. At least, basic metrics such as reliability, CRPS, predictive QQ plot and Brier score should be compared over the conventional method such as PF. Without such evaluation, improvements and features of the hybrid ensemble method cannot be understood in terms of stochastic perspectives.

In addition, Figs 5 and 8 (streamflow hydrographs) should include traces or spreads of ensemble for visual inspection.

3) uncertainty specification on hydrologic DA In order to apply DA for hydrologic modelling, uncertainties for states and observations should be carefully taken care of. Sometimes, not surprisingly, noise configuration or specification may significantly affect DA performance. However, there is no description on how uncertainties of different state variables and observations such as interception, snow, soil moisture and streamflows were formulated and implemented for hydrologic ensemble modelling, which should impact DA process to generate ensemble, optimize state variables and estimate likelihood or weight. A detailed description is required for reproducibility of this study.

Regarding this issue, for example, how different particles of distributed hydrologic models are generated in "the sampling step" of this DA algorithm? More specifically, how high-dimensional model states are being perturbed to avoid sample impoverishment in this step?

4) Under-simulation or filter degeneracy in assimilation step In the analysis or assimilation step which corresponds the first 2-weeks in Figs 5 and 8, under-simulation or filter degeneracy (scenario 3 in Fig. 5 and scenario 2 in Fig. 8) is found. Usually, whatever filter is used, traces of simulated states (here streamflow) overlap observations in the assimilation step since uncertainty of observation is set smaller than that of state variables. It is common that NSE values of the assimilation step or the first forecast step are higher than 0.9 – 0.95. However, a large gap between simulation and observation exists even in the assimilation step, which should be clearly diagnosed and discussed.

5) Comparison of posteriors of state variables What potential readers want to see in the result section may be not only comparison of NSE at the outlet location. The authors need to address why and how their DA method can improve over the conventional ones in hydrologic forecasting from perspectives of distributed modelling. A comparison of posterior distributions of state variables updated by the new and conventional methods may be useful to show how and why the new DA works for high dimensional applications.

Especially, given that the authors urged OPTIMISTS employed essential features from but outperformed particle filters, a comparison of posteriors between two methods is also required to demonstrate whether non-Gaussian and multi-modal distributions are preserved or not.

6) Evaluation and optimization steps for hydrologic modelling It is not clear how the cost function is formulated for distributed hydrologic models. The authors need to show explicitly how multiple spatially-distributed state variables and associated uncertainties are taken into account to formulate the cost function in evaluation and optimization steps.

7) Tuning hyper-parameters There are numerous hyper-parameters such as time step, objectives, no. particles, optimization, Wroot and Kf-class, Psamp and g, related to this DA method which may increase uncertainty and subjectivity of forecasting. However, analysis methods and results on hyper-parameters shown in Figs. 3, 6 and7 are still confusing and do not provide well-organized understandings. A summary or guideline is required for proper range or values of hyper-parameters.

8) Terms and units In Table 3, use of two different units for one variable (m3/s and l/s) is not recommended.

Throughout the manuscript, the term 'time step' is used to represent 'assimilation time step' or 'assimilation window'. Since the time step usually stands for a temporal increment for numerical schemes, 'assimilation window' or 'analysis window' may be more appropriate to avoid possible confusion.

---

## Author Comment (AC3) · 19 Jan 2018

We would first like to express our appreciation to both Referee 1 and Referee 2 for their careful and thorough review of our manuscript. Their comments will certainly help us improve the quality and clarity of our manuscript. Below we offer our response to Referee 1.

**Anonymous Referee 1 on 13 October 2017:** *"I really enjoyed reading the paper, which deals with the important issue of improving model updating techniques for better flood predictions. This manuscript proposes a new data assimilation procedure which combines Bayesian and variational approaches. I believe this is an important contribute*

[Figure]

*to DA research field and of notable interest and modernity, especially for the HESS readership. However, I still have some comments regarding method, structure, and readability of the paper. Below you can find some major comments:"*

We are very glad that you enjoyed the manuscript and found it of value. We appreciate your generous compliments and hope that our work will be a worthy contribution to the community. We thank you for your careful analysis and suggestions. Below we respond to your comments.

*"Results of this research are not well described in the abstract and it is somehow difficult to grasp the main advantage of this approach."*

We will modify the last sentences of the abstract to better convey the findings of our experiments: that OPTIMISTS allowed to produce more robust forecasts when compared to a more traditional method, and that its application on a high-dimensional model was successful in that it maintained the levels of performance observed in the case of lower dimensionality without sacrificing computational efficiency.

*"From the introduction, it is not really clear the difference between OPTIMIST and other hybrid 4D approaches. Novelty has to be better explained in order to further appreciate the added value of such method."*

We will extend the introduction to better convey the differences and our proposed innovations in the revised manuscript. For your information, the main differences were summarized in the conclusions (p16-l28) in the original submission: that OPTIMISTS is inspired in part by the particle filter (instead of being inspired by the EnKF), that it allows for multi-criteria evaluations of candidate particles, and that it utilizes global (evolutionary) optimization methods (instead of using convex ones). Further differences can be extracted from table 1: e.g., that OPTIMISTS allows for non-Gaussian state estimates through kernel density estimation.

*"Nowadays, there are many DA methods with varying complexities and accuracies.*

*However, few of these methods are used in early warning system to improve flood predictions. Did the authors investigate the way to easily implement OPTIMISTS by water authorities for flood forecasting in any existing early warning system? Is there any advantage in terms of computational time if compared to PF and 4D-Var?"*

It is our vision that, once OPTIMISTS' advantages prove greater than its disadvantages (e.g., its relatively higher complexity) in a significant set of test cases, the method will be considered for integration in operational prediction systems for multiple applications. Applicability in high-dimensional cases has been one of our guiding design principles, as we believe this is key for adequate performance in these large-scale/complex systems. While the method currently can be executed in parallel environments and it offers configurations that work well with very large state variable vectors, we are already working in developing enhancements to take the next step in scalability and efficiency, and we will be happy to continue working alongside government agencies or private partners to carry out our vision.

Regarding the comparisons with the PF and 4D-Var, we believe our experiments indicate that OPTIMISTS can provide gains in computational time given that the main strategy for increased performance in most DA methods is through the use of additional model runs: e.g., enlarging the ensemble size in EnKF and PF or the number of candidate solutions explored in the optimization problem in Var. Therefore, showing a more robust performance indicates not only that OPTIMISTS can produce better forecasts with the same resources, but that the same level of performance can be obtained with fewer resources (in this case particles/ simulations).

*"Page 1, lines 14-16: Is this sentence related to the watershed's location or to the use of different models for different case studies? Authors should clarify this point."*

Both: two different locations are used and, because the watersheds are of significantly different scale, we used a different modelling engine for each. We made this decision to diversify the test conditions of our experiments. We will change the wording in this

sentence to convey the correct meaning more clearly.

*"Page 3, lines 8-9: The authors mentioned that "a hybrid data assimilation algorithm that incorporates the most valuable features from both Bayesian and variational methods ". Which ones are these valuable features? Description of Table 1 should be better included within the paper. Right now it looks quite disconnected from the other part of the introduction."*

Our intention was to let Table 1 convey the contrast in features between OPTIMISTS and traditional methods. We will add discussions in the main text so that there is a stronger connection with the table and that these claimed advantages are more easily understood.

*"I found very difficult to follow the flow of thoughts of the authors in describing the DA method. I think it will be beneficial for the readability of the paper to include in section 2 a figure representing the structure of OPTIMISTS. In addition, authors tend to use complex terms for non-DA expert. I suggest revising the description of the paper in order to make it "accessible" to everyone and increase its impact on the scientific community."*

Thank you for your good suggestions. We will use more common terminology in the revised manuscript to accompany technical/domain-specific terms to improve the accessibility of the section. We will also add a figure that will help in understanding how the algorithm works.

*"At this point, results are valid only for the 5 considered flood events and 2 basins obtained. As expected, results largely depend on the features of the flood events and quality of rainfall data. I am afraid that the samll number of events makes results rather random. I suggest to increase the number of flood events to make more general conclusions for this study."*

Thanks for the good suggestions. We will change the comparative test design so that

we can analyze multiple months' worth of forecasts using OPTIMISTS and the particle filter. We already developed some scripts that allow performing assimilation and forecast continuously for this purpose. This will allow for a more thorough and realistic comparison between the two methods. However, we decided to drop the comparison with Evo4DVar as this is not a standard method found in the literature, and its deterministic nature makes a direct comparison complicated.

*"A crucial component in each DA application is the proper definition of model and observational error. While model error is accurately described, I could not see a clear definition of the observations error (standard deviation in Eq.10). The authors have to include more information and references about it."*

We will incorporate the effect of uncertainties in the observations within the description of the algorithm to better convey the multiple ways in which it can be addressed through OPTIMISTS and the contrasts with traditional methods.

*"Are you using actual meteorological forecasts or are you using the observations as perfect forecasts? Please specify"*

In this study, "perfect" meteorological forecasts are used to force the model in all cases (see in page 12, lines 8-13). We also argue that this advantage is applied uniformly to both OPTIMISTS and the other algorithms so that it does not represent an unfair advantage to any. In the revised manuscript, we will make this clearer.

*"I suggest the authors to split results and discussions in two different sections, this would make reading the text so much easier."*

We understand that having separate results and discussion sections allows having the "cold facts" separated from the "subjective" interpretations and opinions of the authors. However, as many authors do, we prefer having these two sections combined because otherwise the discussion section will often have to reference results and figures from the previous section and have the reader jumping back and forth between sections.

This both helps the readability of the article and its compactness. Having both styles be accepted in many communities, we hope this reviewer would agree with us on this point to maintain the two sections combined. We will, however, scan the entire section in search for instances where the distinction between objective results and subjective opinion is not clearly established and revised them accordingly based on this reviewer's good comments.

---

## Author Comment (AC4) · 19 Jan 2018

We would first like to express our appreciation to both Referee 1 and Referee 2 for their careful and thorough review of our manuscript. Their comments will certainly help us improve the quality and clarity of our manuscript. Below we offer our response to Referee 2.

**Anonymous Referee 2 on 21 December 2017:** *"Before beginning review of this manuscript, although not mentioned in the text and reference, this should be considered a re-submission of the previous HESSD manuscript entitled "Hybridizing sequential and variational data assimilation for robust high-resolution hydrologic forecasting*

*(https://doi.org/10.5194/hess-2016-454)" by the same authors, which was rejected in 2016. I suggest the editorial board compare the final revision of the previous HESSD manuscript with the current one if the track record was not screened yet. I cannot examine whether the authors submitted the final revision in the previous submission in 2016 or not. However, if so, improvement and uniqueness of the current manuscript over the rejected final manuscript should be carefully evaluated. In addition, since HESSD is independent publication, the previous manuscript in 2016 should be cited and discussed in this manuscript."*

This point had been discussed with the editor, to whom we expressed that we will adhere to the guidelines required by the journal and the editorial board. This manuscript is a revised version of the cited one in 2016, which takes into account the comments made by the referees back then and by the previous editor. A detailed account of the changes made was submitted to the journal. The 2016 manuscript was rejected by the editor because he considered that the required modifications deserved a more careful timeframe than the one available for the special issue it was submitted to. No final version of the 2016 manuscript was submitted.

*"This manuscript proposed a hybrid DA method, OPTIMISTS, combining sequential and variational methods, and compared performance of developed methodology over PF and VAR using distributed hydrologic models. The topic is of interest to a wide range of hydrologic modelling community. The strategy of the proposed methodology to leverage different DA approaches, sequential and variational DA, is one of the important trends in recent studies. However, there are major gaps in experimental setup and evaluation, and incomplete reasoning in new methodology which require significant changes before publication. I hope the followings would be helpful to improve the quality of manuscript. 1) Evaluation period and methods In this manuscript, the total evaluation period is 10 weeks (5 cases with a 2-weeks period each): 3 scenarios for 2-weeks forecasts in the Blue River and 2 scenarios for 2-weeks forecasts in the Indiantown Run, not including assimilation period. The evaluation period for hydrologic*

[Figure]

*modelling and data assimilation is usually longer than at least 6-8 months and up to multiple decades. The total 10-weeks forecasts (2-weeks piecewise each) and associated metrics cannot be accepted as a rigorous evaluation. Given that the selected events in the Blue River in the 2016 manuscript are different from those in the current one, there seems to a potential to further increase evaluation period. In Table 3, the authors also mentioned calibration periods are 85 and 20 months, respectively. Considering the availability of observation data, what is the maximum evaluation period for two catchments? Why don't you use the whole or most calibration period for DA evaluation? Was there any reason to use the limited period for evaluation? For the larger domain, the Blue River catchment, is there just one streamflow observation gage over 3,000 square kilometer area? Why don't you assimilate observations in multiple locations to reduce equifinality and overfitting? In this study, evaluation metrics were estimated for the whole 2-weeks forecast period. However, it is more common to evaluate metrics for varying forecast lead times because the impact of updating varies and disappears over time. I highly suggest the evaluation period and method should be reconsidered to qualify a kind of general standard shown in many forecast and DA-related papers: simulating more than several months for each catchment and evaluating metrics for varying forecast lead times."*

Thank you for the good suggestions. We will evaluate our method based on the suggestions here in the revised manuscript. We already extended our scripts to allow running an extended-time data assimilation experiment where assimilation is performed with OPTIMISTS continuously to allow producing multiple time series of forecasts with a fixed lead time for the Blue River. We should be able to produce these forecasts for multiple months in order to analyze the performance of the algorithm. We will similarly develop a script to run the particle filter in the same fashion and to be able to compare its forecasts with those of OPTIMISTS.

On the other hand, this new experimental setup will preclude the comparison with the 4D evolutionary variational algorithm for the reason that we were using the same prior

"particle" ensemble to seed its population that the one used for the other methods. However, given that 4DVar is inherently a deterministic approach, such ensemble will not be able to be updated to be the seed of continuous assimilation periods. In practice, variational methods compensate the lack of an ensemble by performing a guided search of the initial state solution space until convergence, but in this case we consider that the model simulation quota that we are allowing each of the methods will not suffice to reach an optimal solution. Moreover, evolutionary variational methods are rare in the literature (more so in operational settings) and therefore we now consider that the comparison would not be of enough significance. While ideally we would like to compare OPTIMISTS with proven 4DVar methods, these require the linearization of the model's dynamics—which is rare in hydrology, would require an enormous amount of additional work, and it is outside the scope of this paper.

Also, while we will implement this new evaluation scheme for the comparison of OPTIMISTS with the particle filter, we will still maintain the scenario-based design for the analysis of the parameters of the assimilator. Even though not "rigorous," we are confident that the variability in the scenarios considered is enough to differentiate relatively adequate configurations of the algorithm. As discussed in the manuscript, we will conduct future tests for cases where there was not enough statistical evidence to conclude one configuration was better than the other. Extended-time evaluations would, in this case, require an extensive computational budget given the large number of parameter combinations selected.

We agree that assimilating data from additional streamflow gages in the Blue River would allow for improved forecasts but, for the purposes of this manuscript, the assimilation of a single measurement provides a balanced challenge that enables the analysis of the strengths and weaknesses of OPTIMISTS in contrast with other methods, and the determination of an adequate set of parameters. That said, we have actually already worked on testing the algorithm using distributed high-resolution observations in a watershed and look forward to include such analyses in a future publication at its due

time.

*"2) Probabilistic evaluation Although the proposed method is a stochastic approach, probabilistic metrics were not measured and analyzed. At least, basic metrics such as reliability, CRPS, predictive QQ plot and Brier score should be compared over the conventional method such as PF. Without such evaluation, improvements and features of the hybrid ensemble method cannot be understood in terms of stochastic perspectives. In addition, Figs 5 and 8 (streamflow hydrographs) should include traces or spreads of ensemble for visual inspection."*

Indeed, probabilistic evaluation is very important to determine if forecasts are over-confident or under-confident. We will include an evaluation metric that allows comparing the confidence of forecasts between OPTIMISTS and the particle filter. We will also modify the plots to show the temporal evolution of the distributions in the revised manuscript.

*"3) uncertainty specification on hydrologic DA In order to apply DA for hydrologic modelling, uncertainties for states and observations should be carefully taken care of. Sometimes, not surprisingly, noise configuration or specification may significantly affect DA performance. However, there is no description on how uncertainties of different state variables and observations such as interception, snow, soil moisture and streamflows were formulated and implemented for hydrologic ensemble modelling, which should impact DA process to generate ensemble, optimize state variables and estimate likelihood or weight. A detailed description is required for reproducibility of this study. Regarding this issue, for example, how different particles of distributed hydrologic models are generated in "the sampling step" of this DA algorithm? More specifically, how high-dimensional model states are being perturbed to avoid sample impoverishment in this step?"*

As explained in subsubsection 2.1.4 in the original manuscript, any numerical objective can be used to judge candidate particles in OPTIMISTS. The likelihood of simulated

outputs given the distributions of the corresponding observations is cited as an example. In such a case, the user would require specifying how the likelihood is computed based on how the error of the observations is being modeled. However, the error metric used in our tests (the mean absolute error, page 12, line 4) is a deterministic one. While this constitutes a departure from the Bayesian theoretical framework, the estimation performed in OPTIMISTS retains its probabilistic character due to the way in which samples are generated and, especially, due to the proposed probabilistic interpretation of the resulting Pareto front (page 7).

Uncertainties in the state variables are all captured by the use of kernel density probability distributions, which is the whole focus of subsection 2.2. The details of the implementation are not introduced in subsection 2.1 but saved for subsection 2.2 because OPTIMISTS offers a modular design in which any type of non-parametric (ensemble-based) probabilistic representation could be used. How new samples are generated from the prior distributions (the core mechanism to "perturb" the ensemble) and how the likelihood of samples given these priors is computed is all explained in this part of the manuscript. While this arrangement was announced in page 4, lines 12-15, we will add reminders on subsubsections 2.1.2 and 2.1.4 in the revised manuscript to make the presentation clearer.

*"4) Under-simulation or filter degeneracy in assimilation step In the analysis or assimilation step which corresponds the first 2-weeks in Figs 5 and 8, under-simulation or filter degeneracy (scenario 3 in Fig. 5 and scenario 2 in Fig. 8) is found. Usually, whatever filter is used, traces of simulated states (here streamflow) overlap observations in the assimilation step since uncertainty of observation is set smaller than that of state variables. It is common that NSE values of the assimilation step or the first forecast step are higher than 0.9 – 0.95. However, a large gap between simulation and observation exists even in the assimilation step, which should be clearly diagnosed and discussed."*

There are several reasons that might explain the relatively low level of agreement seen

between the observations and the adjusted ensemble during the assimilation period. In the first place, it must be noted that the models do indeed have considerable errors, probably mainly in their structures, that prevent them from faithfully replicating the observations at every time step precisely. This is especially apparent in Scenario 3 for the Blue River and both scenarios for Indiantown Run, in which there appears to be conflict between fitting the peaks and fitting the drier inter-peak periods. While both models underwent parameter calibration processes, as documented in subsection 3.1 and in Table 3, no attempt was made to optimize the models' structures (e.g., equations, missing phenomena, resolution, connectivity, etc.). The calibration process, similar to the assimilation, was based on multiple objectives and not only on the maximization of the NSE: we also used the relative error which is more sensitive to errors during dry periods than those during peaks. There is also a telescopic effect of the NSE, according to which, computing it over long periods of time yields higher values than when computed over short ones: for example, if the Indiantown Run model had an overall NSE of 0.81 during the entire calibration period, zooming in on a specific month would result, in average, on a reduced rating. This effect is compounded with the relatively short period of time used for assimilating data and performing forecasts. Finally, a comparably "poor" performance during the assimilation period was also observed for the particle filter and the variational algorithm. With these, we do not find the results to be especially concerning in this regard and, on the other hand, consider that all the provided contrasts are valid given that these conditions were uniform in all cases. In fact, these "defects" reflect the current state-of-the-art challenges in the operational forecasts and it is one of the objectives that we all try to improve from different aspects/angles. We will, however, include a few words in the manuscript regarding these low fitting scores.

*"5) Comparison of posteriors of state variables What potential readers want to see in the result section may be not only comparison of NSE at the outlet location. The authors need to address why and how their DA method can improve over the conventional ones in hydrologic forecasting from perspectives of distributed modelling. A compar-*

*ison of posterior distributions of state variables updated by the new and conventional methods may be useful to show how and why the new DA works for high dimensional applications. Especially, given that the authors urged OPTIMISTS employed essential features from but outperformed particle filters, a comparison of posteriors between two methods is also required to demonstrate whether non-Gaussian and multi-modal distributions are preserved or not."*

We will include probabilistic time series of average soil moisture for forecasts produced both by OPTIMISTS and the particle filter and perform the corresponding analysis. However, we plan to perform detailed analyses of OPTIMISTS' capability of estimating soil moisture, and not only aggregated outputs like streamflow, in a later investigation (when such observations are available). For this study, due to the data limitations at the test watersheds and the length of the manuscript, the distributed comparisons won't be carried.

*"6) Evaluation and optimization steps for hydrologic modelling It is not clear how the cost function is formulated for distributed hydrologic models. The authors need to show explicitly how multiple spatially-distributed state variables and associated uncertainties are taken into account to formulate the cost function in evaluation and optimization steps."*

As explained in page 12, lines 4-7 in the original manuscript, one or two objective functions were used for our experiments: the mean absolute error given the streamflow observations and the likelihood of the particle given the prior state distribution. These objectives can be seen as analogous to the "cost function" used in variational data assimilation, and their equivalence is established in subsubsection 2.1.4. The likelihood is computed using either equation 8 or 9 depending on which type of kernels are used for the state variable distribution. These distributions encode the spatial variability and relationships between state variables in all cells of the model, so the likelihood is thus a measure of how well a candidate particle conforms to the values and (spatial) patterns in the prior distribution. Again due to the limitations of spatial data availability, such

evaluations are not directly carried out in this study in a spatially distributed fashion, but indirectly evaluated through the integrated quantity of streamflow.

*"7) Tuning hyper-parameters There are numerous hyper-parameters such as time step, objectives, no. particles, optimization, Wroot and Kf-class, Psamp and g, related to this DA method which may increase uncertainty and subjectivity of forecasting. However, analysis methods and results on hyper-parameters shown in Figs. 3, 6 and7 are still confusing and do not provide well-organized understandings. A summary or guideline is required for proper range or values of hyper-parameters."*

We acknowledge that using factorial experiments is not a common practice when evaluating the hyper-parameters of these kind of methods. We will revise our presentation of the results to attempt to convey their significance in a more understandable and clearer way. This will include the suggested summary of guidelines for potential OPTIMISTS users to parameterize the algorithm to better fit the needs of their specific application. We will possibly remove Figure 4 which introduces a format different from the other boxplots but that does not provide many significant insights.

*"8) Terms and units In Table 3, use of two different units for one variable (m3/s and l/s) is not recommended."*

We will change the table to use unified units to m3/s.

*"Throughout the manuscript, the term 'time step' is used to represent 'assimilation time step' or 'assimilation window'. Since the time step usually stands for a temporal increment for numerical schemes, 'assimilation window' or 'analysis window' may be more appropriate to avoid possible confusion."*

It is true that the term "assimilation time step" might be confused with the model time step, but it is still necessary to use it because the "assimilation window" or the "analysis window" would refer to the entire period of time in which data is assimilated prior to performing a forecast. As explained in the original manuscript, OPTIMISTS allows

dividing this time window into "time steps" of arbitrary length at which the main loop of the algorithm is executed. These time steps can be as short as the model time step ("sequential," like with particle filters) or as long as the assimilation window (like in 4DVar). The assimilation or analysis window would correspond to two weeks in most of our examples, and the assimilation time step varies from one day to two weeks in the Blue River case or from one hour to four weeks in the Indiantown Run case. We will review the entire manuscript in search for places in which the distinction between model and assimilation time steps could be made more apparent to avoid confusion.

---

## Editor Comment (EC1) · D. Solomatine (Editor) · 22 Jan 2018

The paper has been given a considerable review by two experts in the field. The replies of the authors show that they understand the comments well, and are ready to deal with them, and to revise the manuscript accordingly.

I would stress the importance of citing the paper in HESS-D in 2016 (of the same authors) which is basically the initial version of this manuscript, and explaining whhat is dfferent in the new version. to be submitted to HESS.

The auhtors are encouraged to revise the manuscript and, with the new detailed, point-

by-point replies to the reviews in the Interactive Discussions, to submit it. (Please indicate clearly what is changed in the revised manuscript w.r.t. the original version.)

---

## Author Response (AR1)

**Hybridizing Bayesian and variational data assimilation for robust high-resolution hydrologic forecasting**

Felipe Hernández, Xu Liang

Civil and Environmental Engineering Department, University of Pittsburgh, Pittsburgh, 15213, United States of America

5 Correspondence to: Xu Liang (xuliang@pitt.edu)

**List of changes to the manuscript**

We would like to once again thank the two anonymous referees and the editor for their valuable comments and suggestions which lead to significant modifications to the manuscript. We believe implementing their comments and suggestions has improved the presentation, clearness, and accuracy of our work. Below is the list of changes made in the revision. Specific pages and lines on the modified manuscript are referenced by page and line number as: (p<page number>, llnumber(s)>).

- (p1, 114-16) clarify that two different case studies were used, each with a different modelling engine and a different spatial resolution.
- (p1, 116-19) and (p1, 121-24) summarize the new comparison results in light of the addition of the continuous forecasting experiment and summarize the advantages of OPTIMISTS over existing methods.
- The last two paragraphs of the introduction (p3, 19-28) and the first paragraph of section 2 (p3-4) now include the differences between OPTIMISTS and other hybrid data assimilation approaches.
  - The last two paragraphs of the introduction (p3, 19-28) explain some of the contrasts presented in Table 1 to improve the ties between the text and the table.
  - Figure 1 was expanded to illustrate the entire functional structure of the algorithm and the first paragraph in subsection 2.1 (p4) ties the presented structure with the text.
  - References to subsection 2.2, which deals with the computations with the state probability distribution, are found in subsubsections 2.1.2 (p5) and 2.1.4 (p6) to remind the reader that the details regarding the distributions will be discussed at a later point.
  - The fifth paragraph of subsection 3.1 (p12, 114-23) summarizes how the factorial experiments were set up and its purpose in determining the best assignments to OPTIMISTS parameters.
  - The second-to-last paragraph in subsection 3.1 (p12, l24-29) discusses how OPTIMISTS deals with observation errors in contrast with well-known methods.
  - The original Figure 4 was removed to avoid the confusing details of the analysis of the results of Experiment 1. The main take-away is summarized in (p15, 114-17).

25

20

- Table 5 and Figure 4 show the results of the newly-included continuous forecasting experiment that compares OPTIMISTS and a particle filter in a year-long time frame. The experimental setup is described in subsection 3.2 (p12) and the results are discussed in the last two paragraphs of subsection 4.1 (p15, 118-32).
- Table 5 includes the probabilistic CRPS forecast performance metric in addition to the previous three deterministic performance metrics.
- Figure 4 shows the distribution of the streamflow forecast and not only its mean.
- (p16, 130-p17, 12) discuss why the errors in the scenario-based experiments seem large.
- A section named "Previous versions" was added before the references (p20) that references the 2016 version of the manuscript published in HESSD and mentions the differences between that version and the current one.

[revised manuscript text omitted]

5 where sj represents the jth element of state vector s, and si,j represents the jth element of the ith sample of probability distribution S. Independent/marginal random sampling of each variable can also be applied to replace Eq. (8)(7) by adding random Gaussian residuals to the elements of the selected root sample sroot. Sparse bandwidth matrices (Friedman et al., 2008; Ghil and Malanotte-Rizzoli, 1991)(Friedman et al., 2008; Ghil and Malanotte Rizzoli, 1991) or low-rank approximations (Bannister, 2008; Ghorbanidehno et al., 2015; Li et al., 2015)(Bannister, 2008; Ghorbanidehno et al., 2015; Li et al., 2015)(Context) of the selected root sample size of the selected root sample size of the selected root sample size of the selected root.

[revised manuscript text omitted]

---

## Author Response (AR2)

**Hybridizing Bayesian and variational data assimilation for robust high-resolution hydrologic forecasting**

Felipe Hernández, Xu Liang

Civil and Environmental Engineering Department, University of Pittsburgh, Pittsburgh, 15213, United States of America

5  *Correspondence to*: Xu Liang (xuliang@pitt.edu)

We would like to again express our appreciation to both Referee #1 and Referee #2 for their review of our manuscript. Their new round of comments certainly helped us improve the quality and clarity of our manuscript. Below we offer our response to both referees.

10  ***Anonymous Referee #1:***

*"I really appreciate the efforts that the authors made to include all the suggestions of the reviewers. I do not have any further comment. I wish the authors all the best with this and future studies."*

Thank you very much for your feedback and your good wishes.

***Anonymous Referee #2:***

*"I appreciate the authors' response and the revised manuscript which extended the evaluation period for the low-resolution domain and included one probabilistic measure. I highly evaluate this unified approach to combine two different data assimilation approaches: sequential and variational methods. However, despite the effort, it is still doubtful that the capability*
20  *and robustness of this new data assimilation approach, OPTIMISTS, are properly demonstrated through numerical experiments. Especially, probabilistic features of this new method are not evaluated, and instability is partly observed in the low flow forecasts.*
*As shown in the extended analysis illustrated in Fig. 4, the forecasts by OPTIMISTS show unstable behaviors compared to the particle filter (PF), although such instability is not captured through statistical measures used in this study fortunately. I don't*
25  *think this instability is evidence to give the new method an edge over traditional approaches (Page 16). Neither common Bayesian nor variational approaches lead to such numerical instability while the authors claimed the simulated result as the combined features of those two approaches."*

We agree. After a further investigation, we found that the instabilities occurred for two main reasons: First, it was mostly a
30  problem related to running the testing script in parallel which resulted in the VIC executable failing to finish the simulation in some instances and leading to the corresponding particle to have null results (i.e., to a type of sample impoverishment). This

is a problem with the testing environment and not the algorithm. Second, it was partly produced by the specific selection of OPTIMSTS parameters: low values for $w_{root}$ lead to more random samples being created and low values for $g$ lead to particles in trailing fronts to be weighted more prominently.

The first cause was addressed by running the tests sequentially to avoid the executable failing, although this made the test take
5   much longer to complete. Also, safeguards were implemented to attempt re-running failed simulations. The second cause was addressed by trying configurations with different parameter values (e.g., $w_{root}$ and $g$). We found that there was a tradeoff between allowing for some level of "instability" and the scores obtained for the deterministic error measures. In effect, allowing for more "exploration" (and less "exploitation") leads sometimes to better results thanks to ensemble diversity but sometimes to too much diversity (wider spread). The updated results show much improved behavior (see the new version of Figure 4).
10   Also, the figure is now plotted in logarithmic scale in which instabilities in low flow periods would be greatly amplified.

Additionally, to enhance the testing of the proposed method, we included a continuous test for the high-resolution Indiantown Run watershed spanning three months (with an hourly model time step) in which OPTIMISTS and the PF are compared. The results can be seen in the added Figure 7 and Table 6.

15   *"In the meanwhile, an additional question arises about whether or not model ensemble are properly setup before we discuss data assimilation. Ensemble spreads of PF and OPTIMISTS are extremely narrow in both 6-day and 24-day forecasts, which implies the model uncertainty is not properly represented by ensemble. Even if sampling and regularization steps were devised to avoid sample degeneracy in this study, it is highly likely that the space of state variables could remain narrow and sampling from such narrow space may not lead to optimal solutions whatever kernels or pareto methods are used. I suspect that*
20   *inappropriate ensemble setup may be to blame for low performance in overall DA simulations."*

Yes, the spread of the ensembles is very narrow. As we mention in the manuscript, this is probably a product of the tests ignoring uncertainty in the model parameters (parameters are assumed deterministic without any allowed range of variation) and in the forcings (perfect deterministic forcings are assumed). In operational conditions these uncertainties are not to be
25   undermined this way. The reason we decided to assume these variables to be deterministic was to allow the tests to emphasize state variable uncertainty, which is the focus of data assimilation. The ensembles have a considerable initial spread since they are produced using different sets of model parameters, as described in subsection 3.1. Moreover, we want to reiterate that these conditions are applied uniformly to all cases (including the controls and the competing particle filter) and, therefore, do allow for a fair comparison. We have plans to explore the addition of these types of uncertainties in future experiments, but for this
30   article we want to focus on the important aspects of the method and we think that such assumptions would not invalidate the results. Finally, as we discussed above, OPTIMISTS has the flexibility of being configured to produce forecasts with more variance, but such flexibility should be used with caution as some configurations may result in "instabilities".

*"Although the authors claimed OPTIMISTS efficiently produced 'probabilistic' forecasts in the abstract, no evidence is given to prove this claim. CRPS, the only probabilistic measure used in the revised manuscript, just shows the magnitude of bias. Whether or not posterior distributions produced by the OPTIMISTS are appropriate from probabilistic perspectives still remains unproven. It should be also mentioned that non-normality (non-Gaussian feature) is not evaluated because measures used rely on the first two moments (mean and variance). Rather, given 50% ensemble traces in Fig. 4, if the rank histogram (Talagrand diagram) is estimated, U-shaped diagram is expected because the ensemble spread is too small (For perfect probabilistic forecasts, the rank histogram is flat)."*

Agreed. We incorporated a new probabilistic error metric in both comparison tests to fill this gap: the density of the observations given the forecast distribution (see Table 5 and 6). While we did not include the rank histograms in the manuscript, the results are indeed as expected:

[Figure]

We acknowledge that this constitutes evidence that the forecasts being produced are not perfect by any measure. Further improvements to the assimilation methods should seek to produce better estimates. However, the manuscript inquires on the quality of the forecasts produced by the proposed method in relation with those of an existing method under some (challenging) test circumstances. The included density metric is now able to evaluate the comparative performance of both algorithms in probabilistic terms. Please see page 13, lines 20-30 in the revised manuscript on this.

*"Reproducibility of the algorithm is key to hydrologic research. Although the authors claimed details are explained in subsections, I don't think any hydrologic experts can reproduce the data assimilation procedure by following this manuscript. It remains still vague about how to implement this method for high-dimensional hydrologic modeling. If all are explained as*

*the authors claimed, it may be reasonable to think the current implementation procedure is lacking important part to represent model uncertainty and generate spatial diversity of ensemble."*

Reproducibility is indeed of paramount importance. Although OPTIMISTS is more complex than other existing DA methods, we believe that it can be implemented following the description provided in the manuscript. While we do not provide every detail, due to page limit, regarding the underpinnings of the multiple conceptual frameworks that contributed to the algorithm (PFs, variational DA, kernel density probability distributions, multi-objective evolutionary optimization, etc.), ample resources for their understanding are appropriately referenced throughout the explanation. It is possible that the readers get confused if the terminology utilized does not completely match that of their background discipline, but we believe we have tried as much as we can to navigate this multi-disciplinary approach and included enough cross-referencing of terms and ideas to bridge the differences between them.

It is also not specific which components of the algorithm are unclear to the referee. Regarding "model uncertainty" and "ensemble diversity": Uncertainty in OPTIMISTS is addressed through the use of probability distributions, specifically kernel density (KD) ones. KD distributions are like other non-parametric (Monte Carlo) distributions used in the DA literature in that they are mostly defined by a set of particles or members. Dependencies/relationships between the variables are further represented by the bandwidth matrix. As in EnKFs and PFs, the diversity is mostly dictated by the values on each of the members/particles. Additionally, in our experiments model uncertainty is only attributed to the state variables (not parameters nor forcings).

Using OPTIMISTS for high-resolution modeling requires that the most computationally-intensive equations be replaced by simpler approximations. This is explained in subsection 2.3. Other than that, OPTIMISTS is used in the same way as with low-resolution models.

We would gladly address comments regarding the clarity of the explanations if they are made more specific. Additionally, we have decided to share OPTIMISTS in GitHub so that users can replicate the experiments in the manuscript, perform forecasts for the Blue River and Indiantown Run watersheds, or any other watershed that is included using the VIC and DHSVM models.

**Hybridizing Bayesian and variational data assimilation for robust high-resolution hydrologic forecasting**

Felipe Hernández, Xu Liang

Civil and Environmental Engineering Department, University of Pittsburgh, Pittsburgh, 15213, United States of America

*Correspondence to*: Xu Liang (xuliang@pitt.edu)

**List of changes to the manuscript**

Below is the list of changes made in the revision. Specific pages and lines on the modified manuscript are referenced by page and line number as: (p<page number>, l<line number(s)>).

- (p1, l6-22) Abstract modified to reflect the new results from the continuous forecast experiments on both watersheds.
- (p13, l20-30) Introduction of the new probabilistic error metric (density).
- (p15, l30 – p16, l11) Results of the updated continuous forecast experiment on the Blue River.
- (p17, l3-18) Results of the updated continuous forecast experiment on the Indiantown Run.
- (p19, l21-24) Modified conclusions based on the updated continuous forecast experiments.
- (p20, l2-4) Publication of OPTIMISTS software in GitHub.
- (p28) Update of Figure 4 with new results.
- (p31) Incorporation of Figure 7 with results of continuous experiments in the Indiantown Run).
- (p34) Update of Table 5 with new results.
- (p34) Incorporation of Table 6 with results of continuous experiments in the Indiantown Run).

[revised manuscript text omitted]

---

## Author Response (AR5)

**Hybridizing Bayesian and variational data assimilation for high-resolution hydrologic forecasting**

Felipe Hernández, Xu Liang

Civil and Environmental Engineering Department, University of Pittsburgh, Pittsburgh, 15213, United States of America

5   *Correspondence to*: Xu Liang (xuliang@pitt.edu)

**Editor's comments:**

"*Dear authors*

*Thank you for addressing the last comments.*

10   *My suggestion is to remove word "robust" from the title. Reviewer 2 raised this issue, and I agree it may raise a question of readers. This term never appears in teh text. Term "robust" in modelling has a special meaning, and I don't think you can claim that you check or prove "robustness" of the model.*

*I suggest to split the long paragraph spanning across pages 17-18, e.g. putting before "We offer".*"

Dear Professor Solomatine,

Thank you for your efforts towards the editing of our manuscript. We have modified the title by removing the term "robust".
15   We also divided the long paragraph on subsection 4.1 (page 16, lines 3-22) into three. Below is the new version of the manuscript with the changes tracked. Regarding the title, is there anything we need to do to change the title of the manuscript on the journal's records and on the website?

Thank you very much.

[revised manuscript text omitted]